# CXCR4[high] megakaryocytes regulate host-defense immunity against bacterial pathogens

Jin Wang[1†], Jiayi Xie[2,3†], Daosong Wang[3†], Xue Han[2,3†], Minqi Chen[3], Guojun Shi[1*], Linjia Jiang[2*], Meng Zhao[2,3*]

[1]Department of Endocrinology & Metabolism, The Third Affiliated Hospital, Sun Yat-sen University, Guangzhou, China; [2]RNA Biomedical Institute, Sun Yat-sen Memorial Hospital, Sun Yat-sen University, Guangzhou, China; [3]Key Laboratory of Stem Cells and Tissue Engineering, Zhongshan School of Medicine, Sun Yat-sen University, Ministry of Education, Guangzhou, China

**Abstract** Megakaryocytes (MKs) continuously produce platelets to support hemostasis and form a niche for hematopoietic stem cell maintenance in the bone marrow. MKs are also involved in inflammatory responses; however, the mechanism remains poorly understood. Using single-cell sequencing, we identified a CXCR4 highly expressed MK subpopulation, which exhibited both MK-specific and immune characteristics. CXCR4[high] MKs interacted with myeloid cells to promote their migration and stimulate the bacterial phagocytosis of macrophages and neutrophils by producing TNFα and IL-6. CXCR4[high] MKs were also capable of phagocytosis, processing, and presenting antigens to activate T cells. Furthermore, CXCR4[high] MKs also egressed circulation and infiltrated into the spleen, liver, and lung upon bacterial infection. Ablation of MKs suppressed the innate immune response and T cell activation to impair the anti-bacterial effects in mice under the *Listeria monocytogenes* challenge. Using hematopoietic stem/progenitor cell lineage-tracing mouse lines, we show that CXCR4[high] MKs were generated from infection-induced emergency megakaryopoiesis in response to bacterial infection. Overall, we identify the CXCR4[high] MKs, which regulate host-defense immune response against bacterial infection.

**\*For correspondence:**
shigj6@mail.sysu.edu.cn (GS);
jianglj7@mail.sysu.edu.cn (LJ);
zhaom38@mail.sysu.edu.cn (MZ)

[†]These authors contributed equally to this work

**Competing interest:** The authors declare that no competing interests exist.

## Editor's evaluation

The manuscript by Wang and colleagues studies the heterogeneity of megakaryocytes using single-cell RNA-seq and identifies a subpopulation of CXCR4-high megakaryocytes with immune modulatory roles. Functional studies in this paper show that this subpopulation of megakaryocytes promotes bacterial phagocytosis by macrophages and neutrophils. The compelling evidence for these findings in the manuscript and the fundamental significance of identifying mechanisms by which megakaryocyte subpopulations can impact host defense make this work of interest to researchers in the fields of immunology, hematopoiesis and megakaryocyte biology.

## Introduction

Megakaryocytes (MKs) are large and rare hematopoietic cells in the bone marrow, which continually produce platelets to support hemostasis and thrombosis (*Deutsch and Tomer, 2006*). MK progenitors undergo multiple rounds of endomitosis during maturation to achieve polyploidy (*Chang et al., 2007*; *Deutsch and Tomer, 2013*; *Machlus and Italiano, 2013*; *Nagata et al., 1997*; *Patel et al., 2005*). MKs and their progenitors migrate between distinct microenvironments and organs for their proliferation,

maturation, and biological functions (*Avecilla et al., 2004*; *Fuentes et al., 2010*; *Lefrançais et al., 2017*; *Pal et al., 2020*; *Tamura et al., 2016*; *Wang et al., 1998*). Although platelet generation is the prominent role of MKs, emerging evidence suggests that MKs have other biological functions. Mature MKs interact with HSCs and constitute a unique niche to preserve HSC quiescence in the bone marrow (*Bruns et al., 2014*; *Zhao et al., 2014*). MKs also interact with other niche cells, such as osteoblasts (*Dominici et al., 2009*; *Olson et al., 2013*), non-myelinating Schwann cells (*Jiang et al., 2018*; *Yamazaki et al., 2011*), and blood vessels (*Avecilla et al., 2004*; *Saçma et al., 2019*) to further influence the attraction and retention of hematopoietic stem and progenitor cells during homeostasis and stress.

MK-biased hematopoietic stem cells (HSCs) induce emergency megakaryopoiesis to actively generate MKs upon acute inflammation, which can efficiently replenish the platelet loss during inflammatory insult (*Haas et al., 2015*). Studies suggested that MKs might participate in immune responses independent of their platelet generation role (*Cunin and Nigrovic, 2019*). MKs express multiple immune receptors, such as IgG Fc receptors and toll-like receptors (TLRs), enabling them to sense inflammation directly (*Cunin and Nigrovic, 2019*). Mature MKs also express major histocompatibility complex (MHC) to activate antigen-specific CD8$^+$ T cells and enhance CD4$^+$ T cells and Th17 cell responses through stimulating antigen processing (*Finkielsztein et al., 2015*; *Pariser et al., 2021*; *Zufferey et al., 2017*). Furthermore, MKs release multiple cytokines and chemokines to influence immune cells. For example, MKs produce IL-1α and IL-1β to promote arthritis susceptibility in mice resistant to arthritis (*Cunin et al., 2017*) and produce CXCL1 and CXCL2 to promote neutrophil efflux from the bone marrow (*Köhler et al., 2011*). Lung MKs contribute to thrombosis (*Lefrançais et al., 2017*) and, more interestingly, participate in immune responses (*Pariser et al., 2021*), although the relationship between lung MKs and bone marrow circulating MKs (*Nishimura et al., 2015*) remains unexplored. Furthermore, the recent single-cell atlas shows that MKs are heterogeneous and contain subpopulations that express multiple immune genes and are involved in inflammation response (*Liu et al., 2021*; *Pariser et al., 2021*; *Sun et al., 2021*; *Yeung et al., 2020*). Here, by combining scRNA-seq with functional assays, we identified a CXCR4$^{high}$ MK population, which was generated by infection-induced emergency megakaryopoiesis, and stimulated innate immunity against bacterial infection.

## Results

### Single-cell atlas identifies an immune-modulatory subpopulation of MKs

We applied droplet-based scRNA-seq with CD41$^+$ forward scatter (FSC)$^{high}$ bone marrow MKs to explore the MK heterogeneity (*Figure 1A*; *Figure 1—figure supplement 1A,B*). To enrich accurate MKs, we further performed transcriptomic profile analysis in the phenotypically enriched MKs (*Yeung et al., 2020*). Our scRNA-seq successfully detected 5368 high-quality cells (*Figure 1—figure supplement 1B,C*), in which one MK cluster (1712 cells) and six immune cell clusters (3656 cells) were annotated according to their gene profile (*Figure 1—figure supplement 1D-G*) and the alignment with published scRNA-seq data (*Almanzar et al., 2020*; *Hamey et al., 2021*; *Pariser et al., 2021*; *Xie et al., 2020*; *Yeung et al., 2020*). Our annotated MKs were similar to MKs but distinct to immune cells, including myeloid progenitors, basophils, neutrophils, monocytes, dendritic cells, macrophages, B cells, and T cells, in an integrated scRNA-seq analysis platform (*Figure 1—figure supplement 2*). Therefore, we re-clustered the transcriptionally enriched 1712 MKs into five subpopulations, termed MK1 to MK5 (*Figure 1B*; *Figure 1—figure supplement 3A,B*), which were further confirmed by the integrated scRNA-seq analysis platform to rule out the potential immune cell contamination (*Pariser et al., 2021*; *Xie et al., 2020*; *Yeung et al., 2020*; *Figure 1—figure supplement 3C,D*). We noticed that mature MKs with huge sizes were captured at a relatively low rate, potentially due to the limitation in current techniques in cell purification and single-cell preparation (*Liu et al., 2021*; *Sun et al., 2021*).

Enriched signature genes by Gene Ontology exhibited that MK1 and MK2 highly expressed nuclear division, DNA replication and repair genes for endomitosis (*Figure 1C,D*). MK3 enriched blood coagulation and thrombosis genes for platelet generation (*Figure 1C,D*). No signature pathways were enriched in MK4. MK5 enriched cell migration and immune response genes (*Figure 1C,D*; *Figure 1—figure supplement 4A-E*), cytokine, chemokine (*Figure 1E,F*; *Figure 1—figure supplement 4F*), and genes involved in immune cell interaction (*Figure 1G*, *Figure 1—figure supplement 5A*). MK5 also

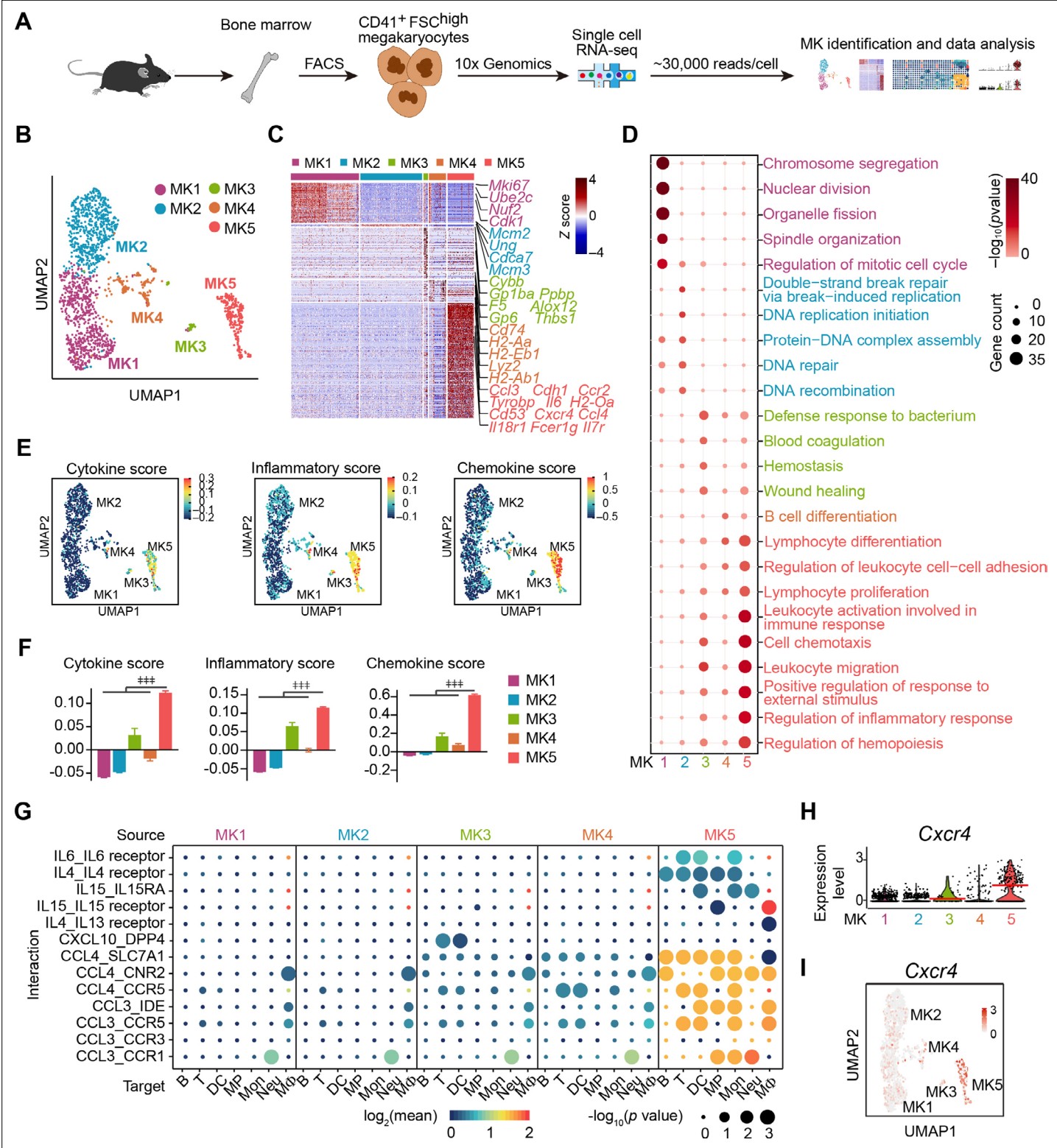

**Figure 1.** Single-cell atlas identifies an immune-modulatory subpopulation of MKs. (**A**) Schematic strategy for MK preparation, scRNA-seq and data analysis. (**B**) Clustering of 1712 bone marrow MKs. (**C**) Heatmap of signature gene expression in MK subpopulations (fold-change >1.5, p value <0.05) with exemplar genes listed on the right (top, color-coded by subpopulations). Columns denote cells; rows denote genes. Z score, row-scaled expression of the signature genes in each subpopulation. (**D**) Gene Ontology (GO) analysis of signature genes (fold-change >1.5, p value <0.05) for each MK subpopulations. GO terms selected with Benjamini–Hochberg-corrected p values <0.05 and colored by –log₁₀(p value). Bubble size indicates the

*Figure 1 continued on next page*

*Figure 1 continued*

enriched gene number of each term. (**E–F**) UMAP visualization (**E**) and statistical analysis (**F**) of cytokine score (left), inflammatory score (middle) and chemokine score (right) in MK1 to 5. (**G**) Dotplots of significant cytokine ligand (source) -receptor (target) interactions between MKs and immune cells discovered. The color indicates the means of the receptor-ligand pairs between two cell types and bubble size indicates p values. Mon, monocytes; MΦ, macrophages (*Dong et al., 2020*); DC, dendritic cells; Neu, neutrophils; MP, myeloid progenitors; T, T cells; B, B cells. (**H–I**) Violin plot (**H**) and feature plot (**I**) of selected signature genes of MK5. Red lines in (**H**) indicate the median gene expression. Repeated-measures one-way ANOVA followed by Dunnett's test for multiple comparisons in (**F**), # <0.01, ## p<0.001.

The online version of this article includes the following source data and figure supplement(s) for figure 1:

**Source data 1.** Signature genes of MK1 to MK5.

**Source data 2.** The means and p value of the average expression level of interacting molecule 1 in cluster 1 and interacting molecule 2 in cluster 2 by CellPhoneDB.

**Figure supplement 1.** Cell isolation, quality control and annotation of scRNA-seq data.

**Figure supplement 2.** Cell type identification by alignment with published scRNA-seq data.

**Figure supplement 3.** Identification of MK subpopulations.

**Figure supplement 4.** Enriched genes in MK1 to 4 and MK5.

**Figure supplement 5.** MK5 interacts with immune cells and express signature genes of immune MKs.

**Figure supplement 6.** Polyploidy, platelet generation ability and cell size of CXCR4$^{low}$ and CXCR4$^{high}$ MKs.

expressed signature genes in recently reported inflammatory-related MKs (*Cd53*, *Lsp1*, *Anxa1*, *Spi*) (*Sun et al., 2021*) and immune MKs (*Ccl3*, *Cd52*, *Selplg*, *Sell*, *Adam8*) (*Liu et al., 2021*; *Figure 1— figure supplement 5B*). We also noticed that MK5 highly expressed *Cxcr4* than other MK subpopulations (*Figure 1H,I*), although most MKs express CXCR4 (*Hamada et al., 1998*; *Figure 1—figure supplement 6A*). To confirm this, we found that CXCR4$^{high}$ MKs expressed MK markers (*Figure 1— figure supplement 6B*), were mainly polyploid cells (*Figure 1—figure supplement 6C*), and had platelet generation ability (*Figure 1—figure supplement 6D*), although they have relatively low polyploidy (*Figure 1—figure supplement 6E*) and smaller cell size (*Figure 1—figure supplement 6F-H*). CXCR4$^{high}$ MKs generated platelets in lower efficiencies compared to CXCR4$^{low}$ MKs (*Figure 1—figure supplement 6D*), suggesting CXCR4$^{high}$ MKs might be specialized for immune functions. Overall, using scRNA-seq, we identified an MK subpopulation that exhibited both MK-specific and immune transcriptional characteristics.

## CXCR4$^{high}$ MKs enhance myeloid cell mobility and bacterial phagocytosis

As MK5 enriched genes involved in myeloid cell activation (*Figure 1—figure supplement 4E*) and myeloid cell interactions (*Figure 1G*, *Figure 1—figure supplement 5A*), we further explored the role of CXCR4$^{high}$ MKs, which enriched MK5, in regulating myeloid immune cells, in regulating the innate immunity function of myeloid cells against pathogens. We challenged mice with *Listeria* (*L.*) *monocytogenes*, a Gram-positive facultative intracellular bacterium (*Bishop and Hinrichs, 1987*; *Edelson and Unanue, 2000*), which induce myelopoiesis (*Eash et al., 2009*; *Figure 2—figure supplement 1A, B*). Interestingly, we noticed that CXCR4$^{high}$ MKs were more dramatically associated with myeloid cells in the bone marrow of mice 3 days after *L. monocytogenes* infection, which was a significant increase than the association between myeloid cells and CXCR4$^{low}$ MKs or the association between randomly placed myeloid cells and CXCR4$^{high}$ MKs (*Figure 2A,B*). The myeloid cell-CXCR4$^{high}$ MK association (mean distance 15.36 μm) was significantly closer than the myeloid cell-CXCR4$^{low}$ MKs association (*Figure 2C*; mean distance 25.62 μm, p=7.0 × 10$^{-4}$ by KS test), and the association between randomly placed myeloid cells and CXCR4$^{high}$ MKs [35.37 μm, $p$ (μ<15.36)=1.8 × 10$^{-10}$] in the bone marrow of mice 3 days after *L. monocytogenes* infection. Whereas the observed mean distance of myeloid cells to CXCR4$^{low}$ MKs (25.62 μm) is not different from random simulations [27.76 μm, $p$ (μ<25.62)=0.14] (*Figure 2C*). This suggested that the increased association between myeloid cell-CXCR4$^{high}$ MK may not be due to the infection-induced expansion of myeloid cells. Furthermore, we did not observe a significant association between myeloid cells and MKs during homeostasis (*Figure 2—figure supplement 1C,D*). We also noticed that bone marrow myeloid cells were preferably adjacent to the CXCR4$^{high}$ MK-blood vessel intersection in mice 3 days after *L. monocytogenes*

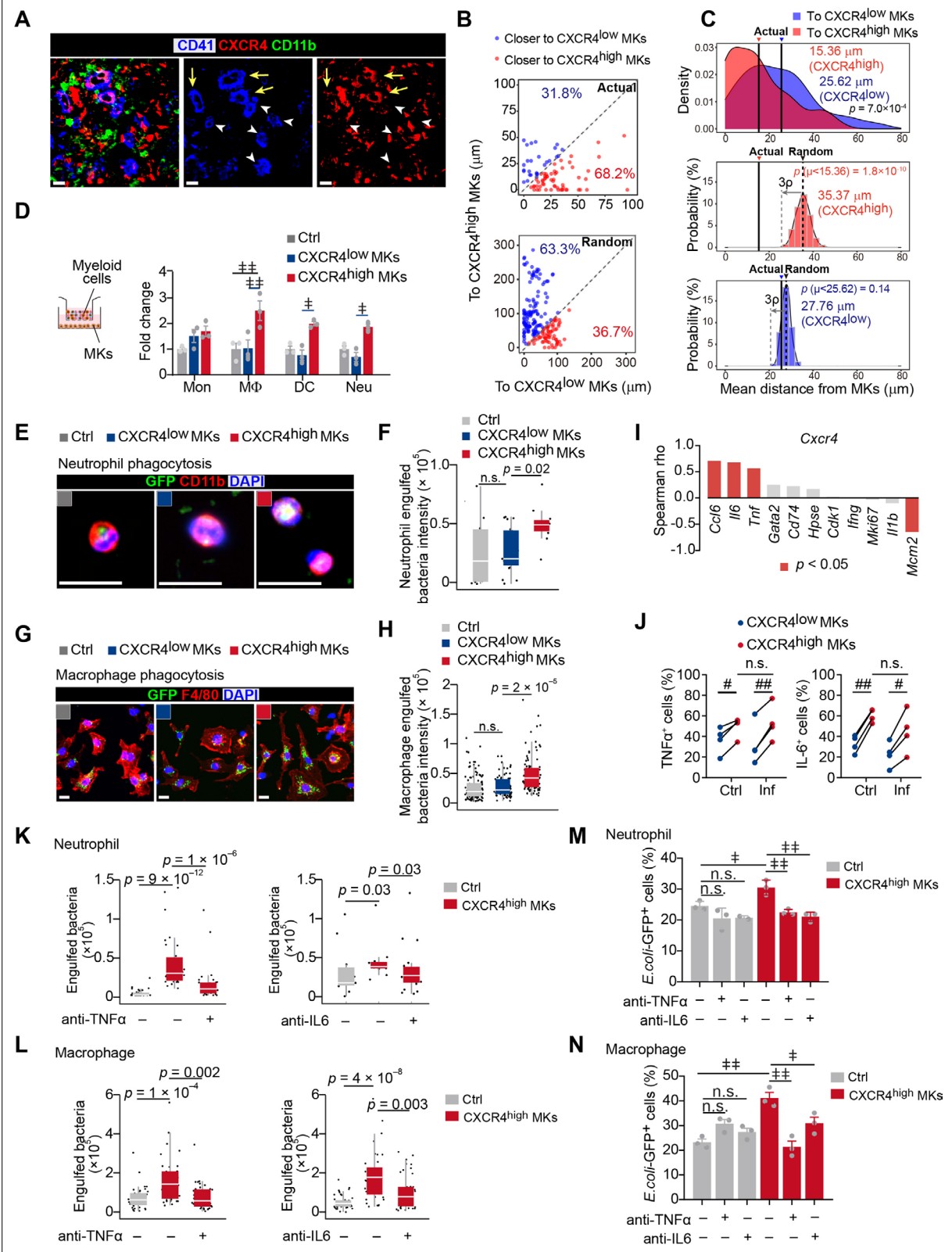

**Figure 2.** CXCR4[high] MKs enhance myeloid cell mobility and bacteria phagocytosis. (**A**) Distribution of myeloid cells to CXCR4[low] or CXCR4[high] MKs 3 days after *L. monocytogenes* infection. Representative images of MKs (blue), CXCR4 (red), and myeloid cells (green) in mouse bone marrow. Yellow arrows indicate CXCR4[high] MKs and white arrowheads indicate CXCR4[low] MKs. (**B–C**) Distance (**B**) and mean distance (**C**) of actual or randomly positioned myeloid cells to the closest CXCR4[low] and CXR4[high] MKs 3 days after *L. monocytogenes* infection. (**D**) Numbers of transmigrated myeloid

*Figure 2 continued on next page*

*Figure 2 continued*

cells normalized to Ctrl (without MKs in the lower chambers) as indicated by transwell assays. Mon, monocytes; MΦ, macrophages; DC, dendritic cells; Neu, neutrophils. (**E–F**) Representative images (**E**) and quantification (**F**) of neutrophil phagocytosis capacity with or without MK co-culture as indicated. CD11b, red; *E. coli*, green; DAPI, blue. Ctrl, neutrophil without MK co-culture. (**G–H**) Representative images (**G**) and quantification (**H**) of macrophage phagocytosis capacity with or without MK co-culture as indicated. F4/80, red; *E. coli*, green; DAPI, blue. Ctrl, macrophage without MK co-culture. (**I**) Spearman correlation analysis between expression profiles of *Cxcr4* and feature genes in MK subpopulations. (**J**) Quantification of TNFα+ and IL-6+ cells in CXCR4low MKs and CXCR4high MKs from control mice or mice 3 days after *L. monocytogenes* infection. (**K–L**) Quantification of neutrophil (**K**) and macrophage (**L**) phagocytosis with or without CXCR4high MK co-culture in the absence or presence of anti-TNFα or anti-IL-6 neutralizing antibodies. (**M–N**) Quantification of the phagocytosis abilities by neutrophils (**M**) and macrophages (**N**) with or without CXCR4low MK or CXCR4high MK co-culture in the absence or presence of anti-TNFα or anti-IL-6 neutralizing antibodies by flow cytometry. Ctrl, neutrophils (**M**) or macrophages (**N**) without MKs co-culture. Scale bars, 20 μm (**A, E, G**). Data represent mean ± s.e.m (**D**) and boxplots show medians, first and third quartiles (**F, H, K–L**). Repeated-measures one-way ANOVA followed by Dunnett's test for multiple comparisons in (**D, M, N**), ‡ $p < 0.05$, ♯ $p < 0.01$, n.s., not significant. A two-sample KS test was performed to assess statistically significant (**C, F, H, K, L**), n.s., not significant. Paired Student's *t*-test was performed to assess statistical significance (**J**), ♯ $p < 0.05$, ♯♯ $p < 0.01$, n.s., not significant.

The online version of this article includes the following source data and figure supplement(s) for figure 2:

**Source data 1.** Distance of actual myeloid cells to the closest CXCR4low and CXR4high MKs 3 days after *L. monocytogenes* infection.

**Source data 2.** Distance of randomly positioned myeloid cells to the closest CXCR4low and CXR4high MKs 3 days after *L. monocytogenes* infection.

**Source data 3.** Mean distance of randomly positioned myeloid cells to the closest CXCR4low and CXR4high MKs 3 days after *L. monocytogenes* infection from 500 times simulations.

**Figure supplement 1.** *L. monocytogenes* promote myelopoiesis and the association of myeloid cells and the CXCR4high MK-blood vessel intersection.

**Figure supplement 2.** CXCR4high MKs promote myeloid cell phagocytosis and produce TNFα and IL-6.

infection (*Figure 2—figure supplement 1E,F*). These observations indicated that CXCR4high MKs might regulate myeloid cells upon bacterial infection.

To explore how CXCR4high MKs regulate myeloid cells, we interestingly found that CXCR4high MKs, but not CXCR4low MKs, effectively promoted myeloid cell mobilization in our transwell assays (*Figure 2D*). Furthermore, we asked whether CXCR4high MKs regulate myeloid cell function against pathogens. To this aim, we incubated purified CXCR4low MKs and CXCR4high MKs with neutrophils or macrophages for bacterial phagocytosis analysis. We found that CXCR4high MKs, but not CXCR4low MKs, efficiently enhanced the bacterial phagocytosis of neutrophils and macrophages (*Figure 2E–H*; *Figure 2—figure supplement 2A,B*).

Our scRNA-seq also exhibited that the high expression of *Cxcr4* was positively correlated with immune cell-stimulating cytokines, such as *Ccl6*, *Tnf*, and *Il6* (*Li et al., 2018*; *Rothe et al., 1993*; *Shapouri-Moghaddam et al., 2018*) in MKs (*Figure 2I*). In line with this, CXCR4high MKs had higher TNFα and IL-6 protein levels than CXCR4low MKs (*Figure 2J*; *Figure 2—figure supplement 2C,D*). The TNFα and IL-6 levels in CXCR4high MKs were comparable to macrophages from mice 3 days after *L. monocytogenes* infection (*Figure 2—figure supplement 2E*), which are known as the primary cellular source of TNFα and IL-6 upon infection (*Shapouri-Moghaddam et al., 2018*). These observations suggested that CXCR4high MKs might stimulate myeloid cell phagocytosis by producing TNFα and IL-6. Indeed, anti-TNFα and anti-IL-6 blocking antibodies compromised the role of CXCR4high MKs in stimulating bacterial phagocytosis of neutrophils and macrophages (*Figure 2K–N*).

## CXCR4high MKs stimulate host-defense immunity against bacterial pathogens

To explore the in vivo role of MKs upon *L. monocytogenes* infection in mice, we employed *Pf4^Cre*; *Rosa26^fs-iDTR* mice, in which MKs were rendered sensitive to diphtheria toxin (DT) (*Zhao et al., 2014*; *Figure 3A and B*). MK ablation increased the number of hematopoietic stem and progenitor cells and myelopoiesis in the bone marrow upon infection (*Figure 3—figure supplement 1A-D*). Notably, MK ablation dramatically increased the bacterial burdens in the liver and spleen 3 days after *L. monocytogenes* infection (*Figure 3C*). We also found that MK ablation reduced the number of myeloid cells, including monocytes, macrophages, dendritic cells (DCs), and neutrophils, in the liver and spleen (*Figure 3D,E*; *Figure 3—figure supplement 1E,F*), suggesting the role of MKs in promoting myeloid cells against pathogens. We further investigated whether MKs regulate adaptative immunity against pathogen infection. Interestingly, we noticed that CXCR4high MKs were able to phagocytose bacteria and presented the ovalbumin (OVA) antigens on their surface via MHC-I (*Figure 3F,G*). Furthermore,

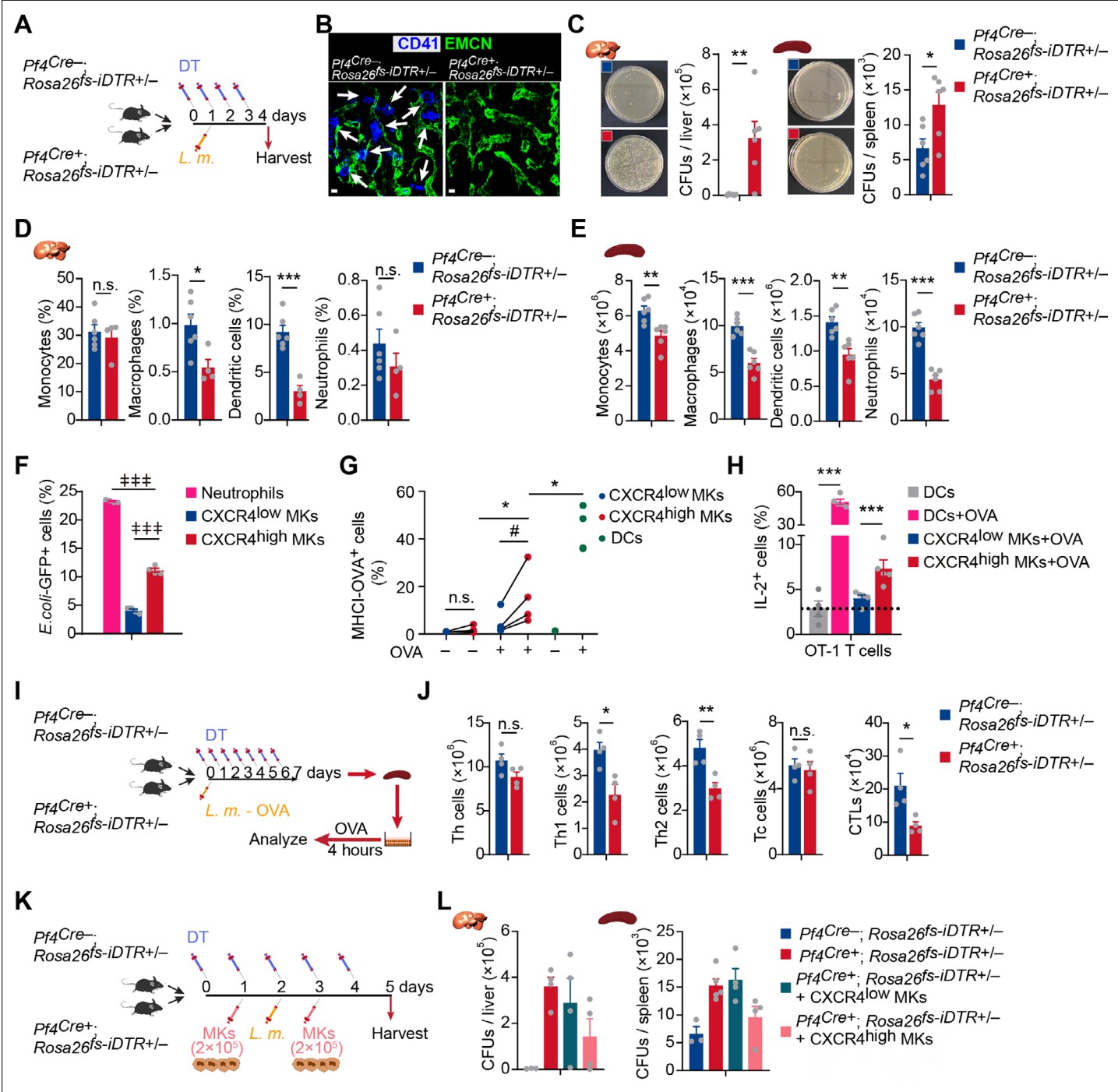

**Figure 3.** CXCR4high MKs stimulate host-defense immunity against bacterial pathogens. (**A**) Schema for diphtheria toxin (DT) and *L. monocytogenes* administration used for the experiments shown in (**B–E**). (**B**) Representative images of MKs (blue, indicated by arrows) and vascular endothelial cells (green) in the bone marrow of mice after four daily DT treatments. (**C**) Bacterial burdens in the liver and spleen of *Pf4Cre; Rosa26fs-iDTR* mice 3 days after *L. monocytogenes* (*L.m.*) infection with four-time DT injections. (**D–E**) Myeloid cells in the liver (**D**) and spleen (**E**) of *Pf4Cre; Rosa26fs-iDTR* mice 3 days after *L. monocytogenes* (*L.m.*) infection with four-time DT injections. (**F**) Quantification of bacterial phagocytosis capacities of neutrophils, CXCR4low MKs and CXCR4high MKs. (**G**) Quantification of MHCI-OVA levels on CXCR4low MKs, CXCR4high MKs and bone-marrow-derived dendritic cells (DCs) upon a pulse of 24 hr with or without OVA. (**H**) Quantification of activated OT-I CD8+ T cells after co-culture with bone-marrow-derived DCs, OVA-pulsed bone-marrow-derived DCs, CXCR4low MKs, or CXCR4high MKs. (**I**) Schema for antigen-specific T cell activation assay shown in (**J**). (**J**) Splenocytes from control or MK ablated mice 7 days after *L. monocytogenes*-OVA$_{257-264}$ infection and seven DT injections were stimulated with OVA peptide in vitro for 4 hr, and antigen-specific activated T cells were quantified (n=4 mice). *L.m.*-OVA, *L. monocytogenes*-OVA$_{257-264}$. (**K**) Schema for DT, *L. monocytogenes* administration, MK transfusing and bacterial burden determination shown in (**L**). (**L**) Bacterial burdens in the liver and spleen of *Pf4Cre; Rosa26fs-iDTR* mice without or with

*Figure 3 continued on next page*

*Figure 3 continued*

CXCR4$^{low}$ or CXCR4$^{high}$ MK transfused. Scale bars, 20μm. Data represent mean ± s.e.m. Repeated-measures one-way ANOVA followed by Dunnett's test for multiple comparisons in (**F**), ⫔ $p < 0.001$. Paired Student's *t*-test was performed to assess statistical significance (**G**), # $p < 0.05$, n.s., not significant. Two-tailed Student's *t*-test was performed to assess statistical significance except (**F**), * $p < 0.05$, ** $p < 0.01$, *** $p < 0.001$, n.s., not significant.

The online version of this article includes the following figure supplement(s) for figure 3:

**Figure supplement 1.** MK ablation influenced HSC and myeloid cell numbers.

**Figure supplement 2.** CXCR4$^{high}$ MKs induced B3Z T cell activation.

OVA antigens presented by CXCR4$^{high}$ MKs activated OT-I CD8$^+$ T cells (*Figure 3H*) and B3Z T cells (*Figure 3—figure supplement 2*), a T cell hybridoma which expresses TCR that specifically recognizes OVA (*Karttunen et al., 1992*). We challenged *Pf4$^{Cre}$; Rosa26$^{fs-iDTR}$* mice with OVA-expressing recombinant microbe (*L. monocytogenes*-OVA). Seven days after *L. monocytogenes*-OVA infection, splenocytes from control or MK ablated mice were re-stimulated with OVA peptide in vitro to assess OVA-specific T cell activation (*Figure 3I*). Notably, MK ablation dramatically reduced the number of CD4$^+$ IFNγ$^+$ Th1, CD4$^+$ IL4$^+$ Th2, and CD8$^+$ cytotoxic T lymphocytes but did not impact the total number of CD4$^+$ T cells and CD8$^+$ T cells (*Figure 3J*). These observations demonstrated that MKs regulate host-defense immunity against *L. monocytogenes* infection. To explore whether CXCR4$^{high}$ MKs contribute to the immune response against bacterial pathogens, we infused the purified CXCR4$^{high}$ MKs and CXCR4$^{low}$ MKs into MK ablation mice during *L. monocytogenes* infection (*Figure 3K*). Notably, we found that the infusion with CXCR4$^{high}$ MKs, but not CXCR4$^{low}$ MKs, partially rescued the bacterial clearance defect in MK ablation mice (*Figure 3L*). This is potentially due to the reduced platelets known for regulating immune responses (*Semple et al., 2011*).

## Bacterial infection stimulates the migration of CXCR4$^{high}$ MKs

High *Cxcr4* expression indicated that CXCR4$^{high}$ MKs might migrate between the bone marrow microenvironment and circulation in response to infection (*Suraneni et al., 2018*). In line with this, our spatial distribution analysis showed that ~80% of MKs directly contacted blood vessels 3 days after *L. monocytogenes* infection, which was much higher than in control mice (~40%) (*Figure 4A,B*; *Figure 4—figure supplement 1A*). Furthermore, more CXCR4$^{high}$ MKs, with small cell sizes (*Figure 1—figure supplement 6F-H*), were tightly associated with blood vessels and trapped in the sinusoid than CXCR4$^{low}$ MKs 3 days after *L. monocytogenes* infection (*Figure 4C,D*). However, *L. monocytogenes* infection did not influence the association between MKs and HSCs (*Figure 4A,E*), albeit the critical role of perivascular MKs in maintaining HSC quiescence (*Bruns et al., 2014*; *Itkin et al., 2016*; *Zhao et al., 2014*) and the dramatic HSC activation upon infection (*Figure 4—figure supplement 1B*).

To further explore the dynamic migration of MKs upon pathogen infection, we adapted an ex vivo real-time imaging method to trace MK migration in the bone marrow (*Xie et al., 2009*). Using *Pf4$^{Cre}$; Rosa26$^{fs-tdTomato}$* mice and ex vivo live imaging approach, we observed that small tdTomato$^+$ MKs rapidly migrated into sinusoids without rupture or platelet release upon infection (*Figure 4F*, *Figure 4—video 1*). In contrast, MKs with large sizes showed much slower migration (*Figure 4—video 1*). Additionally, CXCR4$^{high}$ MKs were decreased in the bone marrow 3 days after *L. monocytogenes* infection but with a similar proliferation and apoptosis rate compared to CXCR4$^{low}$ MKs (*Figure 4G*; *Figure 4—figure supplement 1C-F*), indicating CXCR4$^{high}$ MKs might migrate out of bone marrow. Consistent with this, the frequency of MK5, which enriched CXCR4$^{high}$ MKs, decreased in bone marrow after *L. monocytogenes* infection in our single-cell atlas (*Figure 4—figure supplement 2*). Furthermore, we found that *L. monocytogenes* infection decreased the expression of CXCL12, the ligand of CXCR4 (*Sugiyama et al., 2006*), in bone marrow but increased CXCL12 expression in the lung, liver, and spleen (*Figure 4—figure supplement 3*), suggesting that the distinguished CXCL12 levels between tissues might drive the migration of CXCR4$^{high}$ MKs between tissues. In line with this, CXCR4$^{high}$ MKs were increased in the peripheral blood and organs, including the liver, spleen, and lung 3 days after *L. monocytogenes* infection without an alternation of cell cycle and apoptosis, whereas CXCR4$^{low}$ MKs did not differ except for a slight increase in the liver (*Figure 4G*; *Figure 4—figure supplement 4A-C*). Moreover, inflammatory stresses, such as IFNγ and Lipopolysaccharides (LPS), or *L. monocytogenes* treatment did not increase CXCR4 expression in CXCR4$^{low}$ MKs (*Figure 4H*; *Figure 4—figure supplement 4D*).

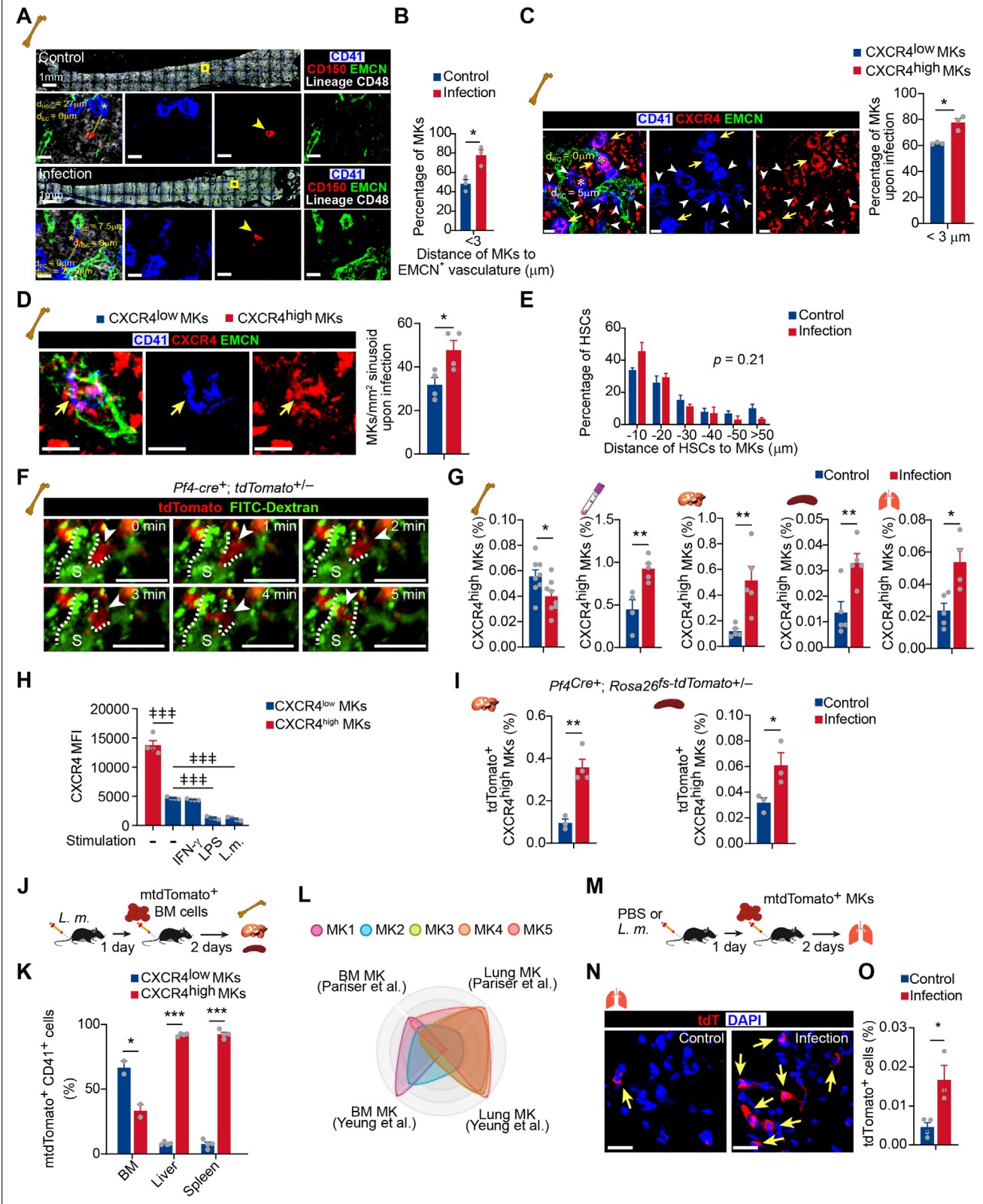

**Figure 4.** Bacterial infection stimulates the migration of CXCR4[high] MKs. (**A**) Representative image of CD41 (blue), CD150 (red), EMCN (green), and lineage cells (white) in bone marrow from control mice or mice at 3 days after *L. monocytogenes* infection. $d_{HSC}$ and $d_{EC}$ indicate the distance between the MK (blue, marked with an asterisk) and the closest HSC (red), endothelial cell (green), respectively. Yellow boxes indicate the locations of the magnified images. Arrowheads indicate HSCs. EMCN, endomucin; EC, endothelial cell. (**B**) Comparison of the distance between MKs to Ecs (n=119

*Figure 4 continued on next page*

*Figure 4 continued*

control and 103 infected MKs) in the bone marrow of control mice or mice at 3 days after *L. monocytogenes* infection. (**C**) Comparison of the distance between CXCR4[low] or CXCR4[high] MKs and endothelial cells (Ecs) in the bone marrow of mice 3 days after *L. monocytogenes* infection (n=68 CXCR4[low] and 78 CXCR4[high] MKs). CD41 (blue), CXCR4 (red), and EMCN (green). Yellow arrows indicate CXCR4[high] MKs, while white arrowheads indicate CXCR4[low] MKs. (**D**) Representative immunofluorescent staining images (left) and quantification (right) of CXCR4 (red) labeled MKs (blue) egressed into sinusoids (green) upon infection (n=46 CXCR4[low] MKs and 69 CXCR4[high] MKs in 4 biological replicates). Yellow arrows indicate CXCR4[high] MKs. (**E**) Comparison of the distance between HSCs to MKs (n=96 control and 127 infected HSCs, p=0.21 by two-sample KS test) in the bone marrow of control mice or mice at 3 days after *L. monocytogenes* infection. (**F**) Visualization of MK migration (red, arrowhead) into sinusoids (green) by live imaging in the bone marrow of *Pf4[Cre+]; Rosa26[fs-tdTomato+/-]* mice 24 hr after *L. monocytogenes* infection (Movie S1). 'S' indicates sinusoid and dashed lines demarcate the border of sinusoids. (**G**) Quantification of CXCR4[high] MKs in bone marrow, peripheral blood, liver, spleen, and lung of control mice and mice 3 days after *L. monocytogenes* infection. (**H**) Quantification of CXCR4 levels on CXCR4[low] MKs treated with IFN-γ, LPS or *L. monocytogenes* for 4 hr compared to CXCR4[high] MKs. (**I**) Quantification of tdTomato[+] CXCR4[high] MKs in the liver and spleen from control mice or mice 3 days after *L. monocytogenes* infection by flow cytometry. (**J**) Schema of mtdTomato[+] bone marrow (from *R26R[mTmG]* mice) cell perfusion in control and *L. monocytogenes* infected recipients. (**K**) The percentage of CXCR4[high] mtdTomato[+] MKs and CXCR4[low] mtdTomato[+] MKs in bone marrow, liver, and spleen of control or infected recipients were analyzed 2 days after mtdTomato[+] bone marrow cells were perfused. (**L**) Radar chart showing transcriptomic similarities of bone marrow MK subpopulations with reported BM and lung MK datasets (*Pariser et al., 2021*; *Yeung et al., 2020*). (**M**) Schema for transfer experiments using tdTomato[+] MKs from *Pf4[Cre+]; Rosa26[fs-tdTomato+/-]* mice into control recipients or recipients 1 day following *L. monocytogenes* infection. (**N–O**) Representative images (**N**) and quantification by flow cytometry (**O**) of tdTomato[+] MKs in the lung of control or infected recipients 2 days after cell perfusion (n=3 mice). Arrows indicate tdTomato[+] MKs in the lung. Scale bars without indicated, 20 μm. Data represent mean ± s.e.m. A two-sample KS test was performed to assess statistical significance in (**E**). Repeated-measures one-way ANOVA followed by Dunnett's test for multiple comparisons in (**H**), # p<0.01, ⧣ p<0.001. Two-tailed Student's *t*-test was performed to assess statistical significance except (**E, H**), * p<0.05, ** p<0.01, *** p<0.001, n.s., not significant.

The online version of this article includes the following video, source data, and figure supplement(s) for figure 4:

**Source data 1.** Distance of HSCs to MKs in the bone marrow from control mice or mice 3 days after *L. monocytogenes* infection.

**Figure supplement 1.** The effects of association of MKs and blood vessels, HSC activation, and MK numbers in bone marrow upon bacterial infection.

**Figure supplement 2.** scRNA-seq of MKs from mice upon bacterial infection.

**Figure supplement 3.** CXCL12 expression upon bacterial infection.

**Figure supplement 4.** Cell cycle and apoptosis of CXCR4[high] MKs, and CXCR4[low] MK numbers in different organs upon bacterial infection.

**Figure supplement 5.** Comparison of immune gene expression in MK5 and lung MKs from control mice or mice upon bacterial infection.

**Figure supplement 6.** Intravascular and extravascular tdTomato[+] MKs in the lung after MK perfusion.

**Figure 4—video 1.** MK migration traced by ex vivo live imaging.

https://elifesciences.org/articles/78662/figures#fig4video1

To further explore how MKs migrate between organs during bacterial infection in vivo, we employed *Pf4[Cre]; Rosa26[fs-tdTomato]*, and *Pf4[Cre]; cell membrane-localized tdTomato cell membrane-localized EGFP (Rosa26[fs-mTmG])* mice in which Tomato or cell membrane-localized EGFP (mGFP) were exclusively expressed in MK lineage (*Tiedt et al., 2007*). mGFP expressing MKs or Tomato expressing CXCR4[high] MKs were increased in the liver and spleen 3 days after *L. monocytogenes* infection (*Figure 4I*; *Figure 4—figure supplement 4E-H*), whereas Tomato expressing CXCR4[low] MKs did not change (*Figure 4—figure supplement 4I*). To further confirm the tissue infiltration of MKs upon infection, we intravenously injected membrane-localized tdTomato (mTomato) expressing bone marrow cells from *Rosa26R[fs-mTmG]* mice into control recipients or recipients infected with *L. monocytogenes* 1 day before mTomato[+] cell perfusion (*Figure 4J*). We found that 2 days after mTomato[+] cell perfusion, engrafted mTomato[+] CXCR4[high] MKs more efficiently infiltrated into the liver (92.1%) and spleen (92.5%); by contrast, most mTomato[+] CXCR4[low] MKs (66.7%) migrated to the bone marrow (*Figure 4K*).

As the lung is an important site for platelet generation (*Lefrançais et al., 2017*), we aligned our MK sc-RNAseq data with lung MKs (*Pariser et al., 2021*; *Yeung et al., 2020*), and found that MK5, MK4, and MK3 showed similar gene profiles with lung MKs (*Figure 4L*). Moreover, MK5 enriched more inflammatory pathway genes, antigen processing, and presentation pathway after *L. monocytogenes* infection, which enabled MK5 to achieve a more similar transcriptional profile as the lung MKs than normal MK5 (*Figure 4—figure supplement 5*). Interestingly, we found that engrafted Tomato[+] MKs (from *Pf4[Cre]; Rosa26[fs-tdTomato]* mice) more efficiently infiltrated the lungs in the infected recipients as extravascular MKs than in the control recipients (*Figure 4M–O* and *Figure 4—figure supplement 6*).

## Acute inflammation-induced emergency megakaryopoiesis generates CXCR4[high] MKs upon infection

Infection-induced emergency megakaryopoiesis compensates the platelet consumption (*Verschoor et al., 2011*). Consistently, we observed that MKs were ruptured in the bone marrow 3 days after *L. monocytogenes* infection to recover the reduced platelets post-*L. monocytogenes* infection (*Couldwell and Machlus, 2019*; *Nishimura et al., 2015*; *Figure 5A–C*). However, CXCR4[high] MKs were increased at 18 hr after *L. monocytogenes* infection and substantially declined at 72 hr in bone marrow, whereas CXCR4[low] MKs remained unchanged upon infection (*Figure 5D*). As MK-committed HSCs drive infection-induced emergency megakaryopoiesis (*Haas et al., 2015*), we asked whether emergency megakaryopoiesis also generates CXCR4[high] MKs to participate in the host-defense response. To this aim, we employed *Scl*[CreER]; *Rosa26*[fs-tdTomato] mice (*Göthert et al., 2005*) to monitor the HSPC derived emergency megakaryopoiesis upon bacterial infection. Eighteen hours after tamoxifen recombing Tomato in HSPCs and *L. monocytogenes* infection (*Figure 5E*), we observed that Tomato[+] HSPCs derived Tomato[+] CXCR4[high] MKs rapidly increased in the bone marrow, similar to the platelet-generating MKs (tdTomato[+] CXCR4[low] MKs) (*Figure 5F*), without a noticeable rise of hematopoietic progenitors (*Figure 5G*). Overall, our observations indicated that CXCR4[high] MKs might be generated by emergency megakaryopoiesis to stimulate pathogen defense.

## Discussion

MKs participate in megakaryocyte maturation, platelet activation, and potentially influence neutrophils and the adaptive immune cells (*Cunin and Nigrovic, 2019*). Accordingly, MKs prevent the spread of dengue virus infection by enhancing the type 1 interferons pathway in murine and clinical biospecimens (*Campbell et al., 2019*) and contribute to cytokine storms in severe COVID-19 patients (*Bernardes et al., 2020*; *Ren et al., 2021*; *Stephenson et al., 2021*). MKs were reported to express multiple inflammation receptors, such as Fcγ receptors (*Markovic et al., 1995*), Toll-like receptors (*Beaulieu et al., 2011*; *Ward et al., 2005*), interleukin receptors (*Navarro et al., 1991*; *Yang et al., 2000*), and IFN receptors (*Negrotto et al., 2011*), which might enable MKs to receive inflammation signals and express cytokines. Recent scRNA-seq studies suggested the existence of MK subpopulations for inflammation responses (*Liu et al., 2021*; *Pariser et al., 2021*; *Sun et al., 2021*; *Wang et al., 2021a*). Here, we identified that MK5 has both MK and immune cell characteristics for platelet generation and immune responses. More importantly, we demonstrated that CXCR4[high] MKs recruited and stimulated innate myeloid cells by producing TNFα and IL-6, for bacterial phagocytosis. Furthermore, CXCR4[high] MKs had the ability for antigen processing and antigen presentation capacity, which suggested that CXCR4[high] MKs might contribute to the regulation of adaptive immune function. This is consistent with a previous observation that lung MKs are able to process and present antigens (*Pariser et al., 2021*). Our data suggested that CXCR4[high] MKs might contribute to the regulation of adaptive immune function. However, as the distinction between CXCR4[high] MKs and CXCR4[low] MKs is not entirely objective, additional markers are warranted to further enrich CXCR4[high] MKs.

We observed that MK ablation increased HSPCS and myeloid granulocyte/macrophage progenitor (GMP) in bone marrow under bacterial infection, which is consistent with 5-FU stress (*Hérault et al., 2017*). However, increased GMP only increased myeloid cells in the bone marrow but not in other organs, which further supported the role of CXCR4[high] MKs in promoting the migration of myeloid cells. Normal HSC to MK development takes 11–12 days in humans and 4 days in mice; However, emergency megakaryopoiesis takes less than a day to generate MKs upon inflammation stress (*Couldwell and Machlus, 2019*; *Liu et al., 2021*; *Sun et al., 2021*; *Figure 5D*). Previously, researchers believed that emergency megakaryopoiesis mainly contributes to the replenishment of damaged platelets upon acute inflammation (*Haas et al., 2015*). We found that inflammation signals could not upregulate CXCR4 in CXCR4[low] MKs in vitro, although we cannot entirely exclude the plasticity of MKs in vivo. Our data showed that CXCR4[high] MKs might be generated from the emergency megakaryopoiesis, instead of CXCR4[low] MKs, to facilitate host-defense responses against bacterial infection.

A recent report showed that the lung is a reservoir of MKs for platelet production (*Lefrançais et al., 2017*). Other works also indicate that lung MKs share a similar transcriptional profile with lung DCs and participate in pathogen infection (*Boilard and Machlus, 2021*; *Pariser et al., 2021*). However, the correspondence between MKs in the lung and bone marrow remains unexplored. Neonatal lung MKs

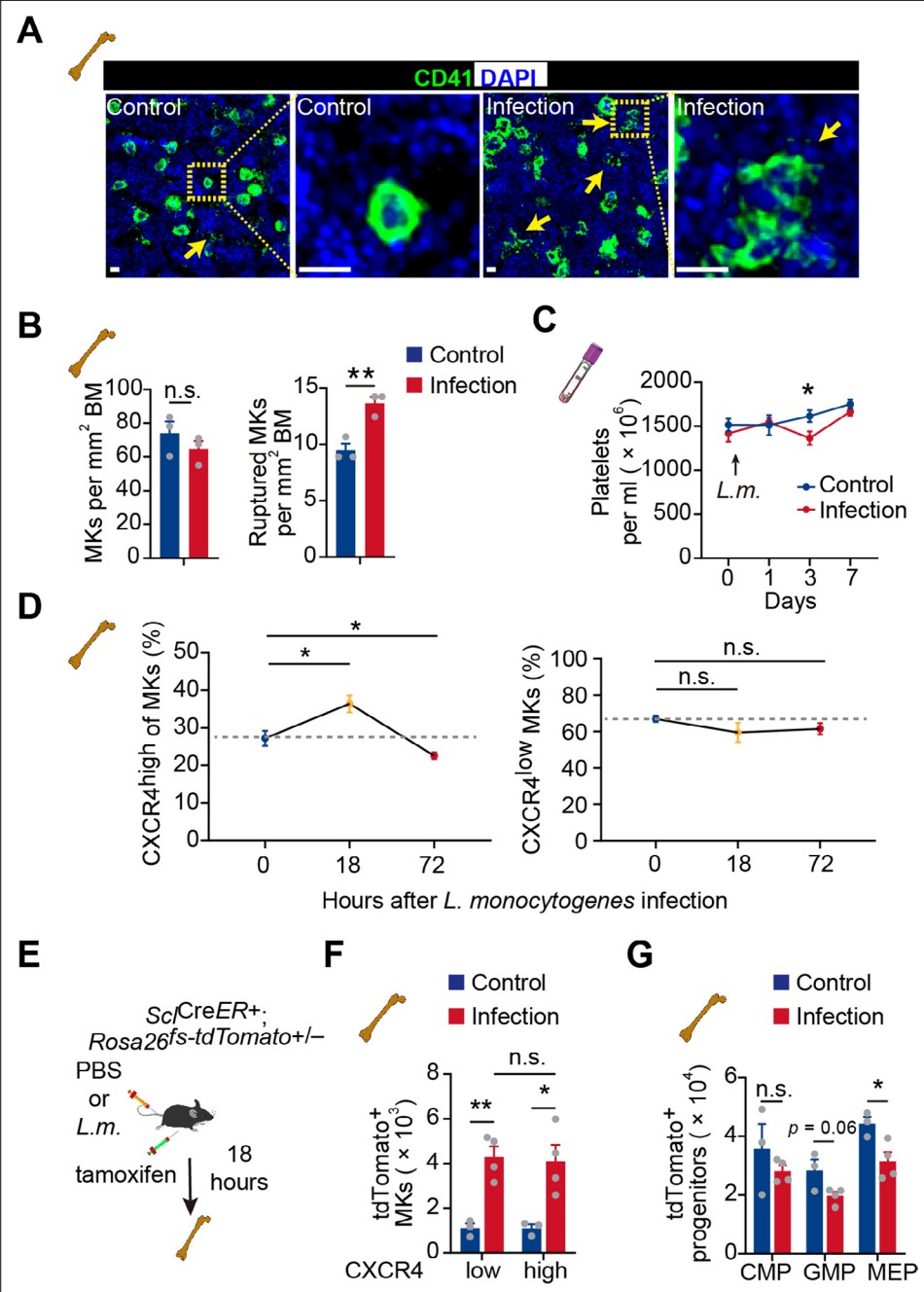

**Figure 5.** Acute inflammation induces emergency megakaryopoiesis of CXCR4^high MKs. (**A–B**) Representative images (**A**) and statistical analysis (**B**) of CD41 (green) and DAPI (blue) in bone marrow from control mice or mice 3 days after *L. monocytogenes* infection. Arrows indicate ruptured MKs, yellow boxes indicate the locations of the magnified images. (**C**) Platelets in peripheral blood in control mice or mice after *L. monocytogenes* infection on indicated days. (**D**) The dynamics percentage of CXCR4^high MKs (left) or CXCR4^low MKs (right) in the bone marrow of *L. monocytogenes*-challenged mice within 72 hr of infection. (**E**) Schema for HSC lineage tracing upon *L. monocytogenes* infection using *Scl^CreER+*; *Rosa26^fs-tdTomato+/-* mice. (**F–G**) Cell numbers of tdTomato^+ CXCR4^low MKs and tdTomato^+ CXCR4^high MKs (**F**), and tdTomato^+ progenitors (**G**) in the bone marrow of control and *L. monocytogenes* infected *Scl^CreER+*; *Rosa26^fs-tdTomato+/-* recipients 18 hr after *L. monocytogenes* infection and tamoxifen administration. CMP, common myeloid progenitor; GMP, granulocyte-monocyte progenitor; MEP, megakaryocyte-erythroid progenitor. Scale bars, 20 μm. Data represent mean ± s.e.m. Two-tailed Student's *t*-test was performed to assess statistical significance, * p<0.05, ** p<0.01, n.s., not significant.

lack the immune molecules in adult lung MKs (*Pariser et al., 2021*), which indicates that lung MKs might have distinct developmental origins. Similarly, MKs are observed to egress and migrate to the pulmonary capillary under stresses (*Davis et al., 1997*). Our works suggested lung MKs might migrate from bone marrow upon infection challenges, although more detailed investigations are warranted in future studies.

# Materials and methods

## Key resources table

| Reagent type (species) or resource | Designation | Source or reference | Identifiers | Additional information |
|---|---|---|---|---|
| Antibody | Anti-CD41a (mouse monoclonal) | eBioscience | Cat#17-0411-82 RRID:AB_1603237 | FACS (1 µl per test) |
| Antibody | Anti-CXCR4 (mouse monoclonal) | eBioscience | Cat#53-9991-80 RRID:AB_953573 | FACS (1 µl per test) |
| Antibody | Anti-CD11b (mouse monoclonal) | eBioscience | Cat#12-0112-82 RRID:AB_2734869 | FACS (1 µl per test) |
| Antibody | Anti-F4/80 (mouse monoclonal) | eBioscience | Cat#17-4801-80 RRID:AB_2784647 | FACS (1 µl per test) |
| Antibody | Anti-Gr-1 (mouse monoclonal) | Biolegend | Cat#108424 RRID:AB_2137485 | FACS (1 µl per test) |
| Antibody | Anti-Ly-6C (mouse monoclonal) | Biolegend | Cat#128022 RRID:AB_10639728 | FACS (1 µl per test) |
| Antibody | Anti-CD11c (mouse monoclonal) | eBioscience | Cat#12-0114-82 RRID:AB_465552 | FACS (1 µl per test) |
| Antibody | Anti-CD45.1 (mouse monoclonal) | eBioscience | Cat#15-0453-82 RRID:AB_468759 | FACS (1 µl per test) |
| Antibody | Anti-CD45.2 (mouse monoclonal) | Biolegend | Cat#109831 RRID:AB_10900256 | FACS (1 µl per test) |
| Antibody | Anti-CD4 (mouse monoclonal) | eBioscience | Cat#12-0041-82 RRID:AB_465506 | FACS (1 µl per test) |
| Antibody | Anti-CD8a (mouse monoclonal) | Biolegend | Cat#100707 RRID:AB_312746 | FACS (1 µl per test) |
| Antibody | Anti-IFN-γ (mouse monoclonal) | Biolegend | Cat#505813 RRID:AB_493312 | FACS (1 µl per test) |
| Antibody | Anti-IL-4 (mouse monoclonal) | Biolegend | Cat#504118 RRID:AB_10898116 | FACS (1 µl per test) |
| Antibody | Anti-CD34 (mouse monoclonal) | eBioscience | Cat#11-0341-82 RRID:AB_465021 | FACS (1 µl per test) |
| Antibody | Anti-Sca-1 (mouse monoclonal) | Biolegend | Cat#108114 RRID:AB_493596 | FACS (1 µl per test) |
| Antibody | Anti-c-Kit (mouse monoclonal) | Biolegend | Cat#105812 RRID:AB_313221 | FACS (1 µl per test) |
| Antibody | Anti-CD135 (mouse monoclonal) | Biolegend | Cat#135314 RRID:AB_2562339 | FACS (1 µl per test) |
| Antibody | Anti-CD3ε (mouse monoclonal) | Biolegend | Cat#100310 RRID:AB_312675 | FACS (1 µl per test) |
| Antibody | Anti-B220 (mouse monoclonal) | Biolegend | Cat#103210 RRID:AB_312995 | FACS (1 µl per test) |
| Antibody | Anti-TER-119 (mouse monoclonal) | Biolegend | Cat#116210 RRID:AB_313711 | FACS (1 µl per test) |
| Antibody | Anti-IgM (mouse monoclonal) | eBioscience | Cat#15-5790-82 RRID:AB_494222 | FACS (1 µl per test) |
| Antibody | Anti-CD16/32 (mouse monoclonal) | Biolegend | Cat#101333 RRID:AB_2563692 | FACS (1 µl per test) |
| Antibody | Anti-CD127 (mouse monoclonal) | Biolegend | Cat#135021 RRID:AB_1937274 | FACS (1 µl per test) |
| Antibody | Anti-TNFα (mouse monoclonal) | Invitrogen | Cat#17-7321-81 RRID:AB_469507 | FACS (1 µl per test) IF (1:100) |
| Antibody | Anti-IL-6 (mouse monoclonal) | Biolegend | Cat#504507 RRID:AB_10694868 | FACS (1 µl per test) IF (1:100) |
| Antibody | Anti-BrdU (mouse monoclonal) | eBioscience | Cat#11-5071-42 RRID:AB_11042627 | FACS (1 µl per test) |
| Antibody | Anti-Endomucin (mouse polyclonal) | R&D | Cat#AF4666 | IF (1:100) |
| Antibody | Anti-CD150 (mouse monoclonal) | Biolegend | Cat#115908 RRID:AB_345278 | IF (1:100) |
| Antibody | Anti-Lineage Panel (mouse monoclonal) | Biolegend | Cat#133307 RRID:AB_11124348 | IF (1:100) |
| Antibody | Anti-Goat AF488 (goat polyclonal) | Invitrogen | Cat#A32814 RRID:AB_2762838 | IF (1:1000) |
| Antibody | Anti-TNF-alpha (mouse monoclonal) | Sino Biological | Cat#50349-R023 | 2 µg ml⁻¹ |

*Continued on next page*

*Continued*

| Reagent type (species) or resource | Designation | Source or reference | Identifiers | Additional information |
|---|---|---|---|---|
| Antibody | Anti-Rabbit AF488 (rabbit polyclonal) | Invitrogen | Cat#R37118 RRID:AB_2556546 | IF (1:1000) |
| Antibody | Anti-OVA257-264 (SIINFEKL) peptide bound to H-2K$^b$ (mouse monoclonal) | Invitrogen | Cat#17-5743-82 RRID:AB_1311286 | FACS (1 µl per test) |
| Antibody | Anti-IL-2 (mouse monoclonal) | eBioscience | Cat#25-7021-80 RRID:AB_1235007 | FACS (1 µl per test) |
| Chemical compound, drug | Diphtheria toxin (DT) | Sigma-Aldrich | Cat#D0564-1MG | 40 µg kg$^{-1}$ body mass |
| Chemical compound, drug | BrdU (5-Bromo-2′-Deoxyuridine) | Sigma-Aldrich | Cat#B5002-250mg | 125 mg kg$^{-1}$ body mass |
| Chemical compound, drug | CFSE (5-Carboxyfluorescein, Succinimidyl Ester) | Invitrogen | Cat#C2210 | 2.5 µM |
| Chemical compound, drug | GM-CSF | Abbkine | Cat#PRP2116 | 10 ng ml$^{-1}$ |
| Chemical compound, drug | IL-4 | novoprotein | Cat#CK15 | 10 ng ml$^{-1}$ |
| Chemical compound, drug | Tamoxifen | Sigma-Aldrich | Cat#T5648 | 20 mg ml$^{-1}$ corn oil |
| Commercial kit | Chromium Single Cell 3′ GEM, Library & Gel Bead Kit v3 | 10 x Genomics | PN-1000075 | |
| Commercial kit | Chromium Chip B Single Cell Kit | 10 x Genomics | PN-1000074 | |
| Cell line (*Mus musculus*) | NCTC clone 929 | ATCC | CCL-1 RRID:CVCL_0462 | |
| Cell line (*Mus musculus*) | B3Z hybridoma CD8 T cell | Dr. Nilabh Shastri | | |
| Other | scRNA sequencing data (raw and processed data) | This paper | GEO: GSE168224 | |
| Genetic reagent (*Mus musculus*) | C57BL/6 J | Shanghai Model Organisms | | |
| Genetic reagent (*Mus musculus*) | Tg(Pf4-icre)Q3Rsko/J (*Pf4$^{Cre}$*) | Jackson Laboratory | Stock No: 008535 | |
| Genetic reagent (*Mus musculus*) | Gt(ROSA) 26Sortm1(HBEGF) Awai/J (*Rosa26$^{fs-iDTR}$*) | Jackson Laboratory | Stock No: 007900 | |
| Genetic reagent (*Mus musculus*) | Gt(ROSA)26Sortm4(ACTB-tdTomato,-EGFP)Luo/J (*Rosa26$^{fs-mTmG}$*) | Jackson Laboratory | Stock No: 007576 | |
| Genetic reagent (*Mus musculus*) | Gt(ROSA)26Sortm9(CAG-tdTomato)Hze/J (*Rosa26$^{fs-tdTomato}$*) | Jackson Laboratory | Stock No: 007905 | |
| Genetic reagent (*Mus musculus*) | *Scl$^{CreER}$* mice | **Göthert et al., 2005** | | |
| Genetic reagent (*Mus musculus*) | Cxcl12tm2.1Sjm/J (*Cxcl12$^{fs-DsRed}$*) | Jackson Laboratory | Stock No: 022458 | |
| Genetic reagent (*Mus musculus*) | C57BL/6-Tg(TcraTcrb)1,100Mjb/J (*OT-I*) | Jackson Laboratory | Stock No: 003831 | |
| Strain, strain background (L. monocytogenes) | 10403 S | **Bishop and Hinrichs, 1987** | | |
| Software, algorithm | Cell ranger_3.0.2 | 10 x Genomics | tenx RRID:SCR_01695 | |
| Software, algorithm | R_3.6.3 | https://cran.r-project.org/ | R 3.6.3 | |

*Continued on next page*

*Continued*

| Reagent type (species) or resource | Designation | Source or reference | Identifiers | Additional information |
|---|---|---|---|---|
| Software, algorithm | Seurat_3.0.2 | *Butler et al., 2018* | Seurat RRID:SCR_016341 | |
| Software, algorithm | ggplot2_3.2.0 | https://cran.r-project.org/web/packages/ggplot2/index.html | ggplot2 RRID:SCR_014601 | |
| Software, algorithm | clusterProfiler_3.12.0 | *Yu et al., 2012* | clusterProfiler RRID:SCR_016884 | |
| Software, algorithm | pheatmap_1.0.12 | https://cran.r-project.org/web/packages/pheatmap/ | pheatmap RRID:SCR_016418 | |
| Software, algorithm | CellPhoneDB_2.1.7 | *Efremova et al., 2020* | CellPhoneDB RRID:SCR_017054 | |
| Software, algorithm | CellChat_1.1.3 | *Jin et al., 2021* | CellChat 1.1.3 | |
| Software, algorithm | symphony_1.0 | *Kang et al., 2021* | symphony 1.0 | |
| Software, algorithm | MetaNeighbor_1.10.0 | *Crow et al., 2018* | MetaNeighbor RRID:SCR_016727 | |
| Software, algorithm | iMAP_1.0.0 | *Wang et al., 2021b* | iMAP 1.0.0 | |
| Software, algorithm | scmap_ 1.16.0 | *Kiselev et al., 2018* | Scmap RRID:SCR_017338 | |
| Software, algorithm | enrichplot_1.4.0 | *Yu, 2019* | enrichplot 1.4.0 | |
| Software, algorithm | Imaris_8.4 | Bitplane | Imaris RRID:SCR_007370 | |
| Software, algorithm | FlowJo_10 | BD Bioscience | FlowJo RRID:SCR_008520 | |
| Software, algorithm | ImageJ_ 1.8.0 | National Institutes of Health | ImageJ RRID:SCR_003070 | |
| Other | DAPI (4',6-Diamidino-2-Phenylindole, Dihydrochloride) | Thermo Fisher | Cat#D1306 | IF (0.5 µg/mL) |
| Other | Corn oil | Sigma-Aldrich | Cat#PHR2897 | Tamoxifen dissolution |
| Other | Lymphocyte Separation Medium | TBD Science | Cat#LTS1077 | Liver cell isolation |

## Mice

C57BL/6-Tg(*Pf4-cre*)Q3Rsko/J (*Pf4$^{Cre}$*), C57BL/6-Gt(ROSA) 26Sortm1(HBEGF) Awai/J (*Rosa26$^{fs-iDTR}$*), C57BL/6-Gt(*ROSA*)26Sortm4(ACTB-tdTomato,-EGFP)Luo/J (*Rosa26$^{fs-mTmG}$*), Gt(ROSA)26Sortm9(CAG-tdTomato)Hze (*Rosa26$^{fs-tdTomato}$*), CXCL12tm2.1Sjm/J (*Cxcl12$^{DsRed}$*) and C57BL/6-Tg(TcraTcrb)1,100Mjb/J (*OT-I*) mice were obtained from the Jackson Laboratory. *Scl$^{CreER}$* mice were provided by J. R. Göthert. All mice were maintained in the C57BL/6 background. Animals were blindly included in the experiments according to genotyping results as a mix of male and female. All animal experiments were performed according to protocols approved by the Sun Yat-sen University animal care and use committee (approval No. SYSU-IACUC-2021-B0617).

## Cell line

B3Z hybridoma T cells were kindly gifted by Dr. Nilabh Shastri (Johns Hopkins University). This cell line was verified to be mycoplasma free by EZdetect PCR Kit for Mycoplasma Detection (HiMedia).

## Bacteria and infections

*Listeria* (*L.*) *monocytogenes* infection was performed as described with minor modifications (*Edelson and Unanue, 2000*; *Verschoor et al., 2011*). In brief, wild-type *L. monocytogenes* strain 10,403 S grown to exponential phase at 37 °C in TSB media was injected intravenously at a dose of 2500 colony-forming units (CFUs) to determine spleen and liver bacterial burdens 3 days after infection. Recombinant *L. monocytogenes* expressing the chicken ovalbumin peptide (OVA$_{257-264}$) (*L.m.* – OVA$_{257-264}$) was injected intravenously at a dose of 2500 CFUs to determine activated spleen T cells 7 days after infection. *Escherichia* (*E.*) *coli* wild-type strain 85,344 expressing GFP was constructed as previously described (*Feng et al., 2020*). GFP-labeled *E. coli* was grown to exponential phase at 37 °C in LB media and washed with PBS before being suspended for phagocytosis assays.

## Antibodies for flow cytometry analysis and cell sorting

For cell sorting and analysis, monoclonal antibodies to CD41 (MWReg30, eBioscience), CXCR4 (2B11, eBioscience), CD11b (M1/70, eBioscience), F4/80 (BM8, eBioscience), Gr-1 (RB6-8C5, Biolegend),

Ly6C (HK1.4, Biolegend), CD11c (N418, eBioscience), CD45.1 (A20, eBioscience), CD45.2 (104, Biolegend), CD4 (GK1.5, eBioscience), CD8 (53–6.7, Biolegend), INF-γ (XMG1.2, Biolegend), IL4 (11B11, Biolegend), CD34 (RAM34, eBioscience), Sca-1 (D7, Biolegend), c-kit (2B8, Biolegend), CD135 (A2F10, Biolegend), CD3ε (145–2 C11, Biolegend), CD45R (RA3-6B2, Biolegend), TER-119 (Ter-119, Biolegend), IgM (II/41, eBioscience), FγRII (93, Biolegend), IL-7R (A7R34, Biolegend), TNFα (MP6-XT22, Invitrogen), IL-6 (MP5-20F3, Biolegend), OVA257-264 (SIINFEKL) peptide bound to H-2K$^b$ (eBio25-D1.16 (25-D1.16), Invitrogen) and IL-2 (JES6-5H4, eBioscience) were used where indicated.

## Flow cytometry and cell sorting

Bone marrow cells were isolated from mouse femora and tibiae as previously reported (*Jiang et al., 2018*). Splenocytes were mechanically dissociated in PBS with 2% FBS. Peripheral blood was collected from the retro-orbital sinus and anticoagulated by K2-EDTA. Those three kinds of cells then underwent red blood cell lysis for 5 min using 0.16 M ammonium chloride solution. Liver cells were mechanically dissociated and lysed using 0.16 M ammonium chloride solution, followed by gradient sedimentation using a density reagent (LTS1077, TBD Science) following the manufacturer's instruction. Cell sorting was performed using a cell sorter (MoFlo Astrios, Beckman Coulter) with a 100 µm nozzle at a speed of around 5000 cells s$^{-1}$. For intracellular cytokine staining, cells were pretreated with Brefeldin-A (BFA, 10 µg ml$^{-1}$) for 4 hr at 37°C before staining. For MK antigen presentation detection, MKs were co-culture with 100 µg ml$^{-1}$ soluble full-length OVA for 24 hr before staining. For IFNγ, LPS and *L. monocytogenes* treatment, cells were co-culture with 10 ng ml$^{-1}$ IFN-γ or 30 µg ml$^{-1}$ LPS for 4, 18, or 24 hr, or 10$^6$ *L. monocytogenes* for 4 hr in a 37°C incubator before staining. Cell analysis was performed on either one of the flow cytometers (Attune NxT, Thermo Fisher; Cytek AURORA, Aurora).

## Single-cell library construction and sequencing

Sorted CD41$^+$ FSC$^{high}$ single cells from four mice of a control MK group and an MK group from mice 3 days upon *L. monocytogenes* infection each were processed through the Chromium Single Cell Platform using the Chromium Single Cell 3' Library and Gel Bead Kit v3 (PN-1000075, 10 x Genomics) and the Chromium Single Cell B Chip Kit (PN-1000074, 10 x Genomics) as the manufacturer's protocol. In brief, over 7000 cells were loaded onto the Chromium instrument to generate single-cell barcoded droplets. Cells were lysed and barcoded reverse transcription of RNA occurred. The library was prepared by following amplification, fragmentation, adaptor, and index attachment then sequenced on an Illumina NovaSeq platform.

## scRNA-seq processing

The scRNA-seq reads were aligned to the mm10 reference genomes, and unique molecular identifier (UMI) counts were obtained by Cell Ranger 3.0.2. Normalization, dimensionality reduction, and clustering were performed with the Seurat 3.0 R package (*Butler et al., 2018*). For the control and *Listeria* (*L.*) *monocytogenes* infection group, we loaded one 10 x Genomics well each and detected 5663 and 5948 cells that passed the Cell Ranger pipeline, respectively. To ruled out low quality cells, cells with >12% mitochondrial content or <200 detected genes were excluded with Seurat function subset (percent.mt <12 & nFeature_RNA >200). We ruled out doublets with default parameters of Doublet-Decon R package, and 54 control cells and 939 *L. monocytogenes* infected cells were excluded. Following the standard procedure in Seurat's pipeline, we identified 3272 MKs from control mice (1712 MKs) and mice with *L. monocytogenes* infection (1560 MKs) (3897 and 3449 immune cells were discarded, respectively) in combination with MetaNeighbor method. Preprocessed dataset normalization was performed by dividing the UMI counts per gene by the total UMI counts in the corresponding cell and log-transforming before scaling and centering. SCT normalization was performed with the script: object <- SCTransform(object, vars.to.regress = ""percent.mt"", verbose = FALSE). Signature genes of each cluster were obtained using the Seurat function FindMarkers with Wilcox test with fold change >1.5 and p value <0.05 after clustering. Heatmaps, individual UMAP plots, and violin plots were generated by the Seurat functions in conjunction with ggplot2 and pheatmap R packages.

Similarities and UMAP projection between our scRNA-seq data and published datasets GSE152574 (*Yeung et al., 2020*), GSE158358 (*Pariser et al., 2021*), GSE137540 (*Xie et al., 2020*), GSE128074 (*Hamey et al., 2021*), or GSE132042 (*Almanzar et al., 2020*) were conducted by MetaNeighbor R package (*Crow et al., 2018*), iMAP.py and Symphony R package (*Kang et al., 2021*). iMAP integration

was performed using the default parameters except n_top_genes = 2000, min_genes = 0, min_cells = 0, and n_epochs = 100 before doing dimensionality reduction using Uniform Manifold Approximation and Projection method (UMAP, n_neighbors = 30, n_pca = 30). Radar charts were generated with JavaScript written by Nadieh Bremer (https://www.visualcinnamon.com/). Euclidean distances denote the distances between the centroid of each cluster.

Correlations were calculated based on normalized RNA values, with the function cor and the parameter 'method = "spearman"'. Multiple testing correction using the function cor.test with the parameter "method = "spearman" and it was applied for *Cxcr4* expression correlations. We calculated the similarities between MK1 to 5 with the published MK, immune cell, and myeloid progenitor datasets (*Almanzar et al., 2020*; *Hamey et al., 2021*; *Pariser et al., 2021*; *Xie et al., 2020*; *Yeung et al., 2020*) using scmap R package (*Kiselev et al., 2018*). Default parameters and 1000 features were used and threshold >0 was set. Cell-type matches are selected based on the highest value of similarities and the second-highest value which is not 0.01 less than the highest value across all reference cell types.

Cytokine, inflammatory, chemokine, and antigen processing and presentation scores were evaluated with the AddModuleScore function of Seurat using genes from KEGG pathway ko04060, cytokine-cytokine receptor interaction; GO:0006954, inflammatory response; chemokine ligands from CellPhoneDB.mouse (*Jin et al., 2021*) and GO:0019882, antigen processing and presentation.

Interaction analysis of MKs and immune cells were conducted by CellPhoneDB (*Efremova et al., 2020*) (transformed to human orthologous genes *Davidson et al., 2020*) and CellChat R package (*Jin et al., 2021*). Only interactions involving cytokines were shown. Gene Ontology (GO) analysis was performed using clusterProfiler R package (*Yu et al., 2012*) and visualized using enrichplot R package (*Yu, 2019*).

Gene set enrichment analysis (GSEA) was performed using gsea R package (*Subramanian et al., 2005*) and visualized using enrichplot R package. Gene lists were pre-ranked by the fold change values of the differential expression analysis using Seurat function FindMarkers. Gene sets for GSEA were obtained from GO database (GO:0002367, cytokine production involved in immune response; GO:0006954, inflammatory response; GO:0008009, chemokine activity; GO:0022409, positive regulation of cell-cell adhesion; GO:0002275, myeloid cell activation involved in immune response).

Gene set variation analysis (GSVA) was performed using GSVA R package (*Hänzelmann et al., 2013*). GSVA was performed to calculate GSVA score of indicated pathway genes in single cell datasets with the whole protein encoding genes after log normalization of expression values. Gene sets for GSVA were obtained from GO database (GO:0022409, positive regulation of cell-cell adhesion; GO:0002275, myeloid cell activation involved in immune response; GO:0002367, cytokine production involved in immune response; GO:0007596, blood coagulation; GO:0019882, antigen processing and presentation; GO:0034340: response to type I interferon; GO:0034341: response to interferon-gamma; GO:0045088, regulation of innate immune response; GO:0042742, defense response to bacterium; GO:0002819, regulation of adaptive immune response; GO:1903708, positive regulation of hemopoiesis).

## Lung cells preparation for flow cytometry

Lungs were removed and digested as described with minor modifications (*Lefrançais et al., 2017*). In brief, removed lungs were placed in 1.5 ml tubes, minced with scissors, and digested with 1 ml digestion buffer (HBSS with 1 mg ml⁻¹ collagenase D, 0.1 mg ml⁻¹ DNase I, 25 mM HEPES, 2 mM L-glutamine, and 2% FBS) for 30 min at 37°C before filtration through a 100 μm cell strainer and red blood cell lysis for 5 min. Samples were then filtered through 70 μm strainers and resuspended for subsequent surface marker staining for flow cytometry.

## Megakaryocyte ablation induction

*Pf4^Cre* mice were mated with the *Rosa26^fs-iDTR* line to generate *Pf4^Cre; Rosa26^fs-iDTR* mice. Diphtheria toxin (DT, Sigma-Aldrich) was injected intraperitoneally every day at a dose of 40 ng g⁻¹ bodyweight into *Pf4^Cre+; Rosa26^fs-iDTR+/−* mice and their cre negative counterparts to induce megakaryocyte ablation as indicated.

## Cre-ER recombinase induction

$Scl^{CreER}$ mice were mated with the $Rosa26^{fs-tdTomato}$ line to generate $Scl^{CreER}$; $Rosa26^{fs-tdTomato}$ mice. For induction of cre-ER recombinase, $Scl^{CreER}$, $Rosa26^{fs-tdTomato+/-}$ mice were injected with tamoxifen intraperitoneally once (2 mg in 0.1 ml corn oil; Sigma-Aldrich).

## BrdU incorporation assay

5-Bromo-2-deoxyuridine (BrdU) was administered at a single dose of 125 mg kg$^{-1}$ body mass by intraperitoneal injection. Whole bone marrow cells were collected 12 hr later and incubated with anti-CD41 and anti-CXCR4 for 1 hr. Cells were washed and then fixed with 4% PFA at 4 °C overnight. Cells were then permeabilized with 0.5% TritonX-100 for 15 min at room temperature and incubated with 1 mg ml$^{-1}$ DNase I (Roche) for 1 hr at 37 °C followed by incubating with anti-BrdU (BU20A, eBioscience) for 1 hr at room temperature before being analyzed.

## Annexin V binding assay

For Annexin V binding assay, bone marrow cells were incubated with cell surface markers for 1 hr at 4 °C and then washed with PBS before being resuspended with Annexin V binding buffer (Biolegend). Cells were then incubated with FITC Annexin V (Biolegend) for 15 min at room temperature in dark, and then 300 µl Annexin V binding buffer was added to each tube. Cells were analyzed by a flow cytometer.

## Immunostaining

Immunostaining of frozen sections was performed as described (*Jiang et al., 2018*). For bone sections, mice were perfused with PBS and 4% paraformaldehyde (PFA). Then the bones were fixed with 4% PFA for 24 hr, decalcified with 0.5 M EDTA for 2 days, and gradient dehydrated by 15% and 30% sucrose for another 2 days. The thick of sections was 30 µm. We used CD41 (MWReg30; eBioscience; 1:200), Endomucin (R&D; 1:100), CD150 (TC15-12F12.2; Biolegend; 1:100), CD48 (HM48-1; Biolegend; 1:100), CXCR4 (2B11, eBioscience; 1:100) antibodies, and lineage panel (Biolegend; cat #133307; 1:50). Secondary staining was done with donkey anti–goat AlexaFluor 488 (Invitrogen; 1:1000). For the liver and spleen from $Pf4^{Cre+}$; $Rosa26^{fs-mTmG+/-}$ mice, and lung from $Pf4-cre^{+}$; $Rosa26^{fs-tdTomato+/-}$ mice, we used DAPI (Thermo Fisher; 0.5 µg ml$^{-1}$) to stain the frozen sections. For phagocytosis analysis, F4/80 (BM8, eBioscience; 1:100), CD11b (M1/70; Invitrogen; 1:100), CD41 (MWReg30; Thermo Fisher; 1:200) and DAPI was used. For sorted MKs, we used CXCR4 (2B11, eBioscience; 1:100), TNFα (R023, Sino Biological; 1:100) and IL-6 (MP5-20F3, Biolegend; 1:100) antibody. Secondary staining was performed with donkey anti-rabbit AlexaFluor 488 (Invitrogen; 1:1000). Confocal images were obtained using a spinning-disk confocal microscope (Dragonfly, Andor) and analyzed using Imaris 9.0 software (Oxford Instruments). Three-Dimension plots were generated using Matplotlib (*Hunter, 2007*).

## Quantitative real-time (qRT-) PCR

For RT-qPCR, MKs were dissociated in Trizol (Magen), and RNA was extracted following the manufacture's instruction. RNA was reverse transcribed into cDNA using the TransCript All-in-One First-Strand cDNA Synthesis kit (Transgene). Quantitative PCR was performed using a Bio-Rad CFX 96 touch. The primers for *Pf4* were 5'-GGGATCCATCTTAAGCACATCAC-3' (forward) and 5'-CCATTCTTCAGG GTGGCTATG-3' (reverse). The primers for *Vwf* were 5'-CTTCTGTACGCCTCAGCTATG-3' (forward) and 5'-GCCGTTGTAATTCCCACACAAG-3' (reverse). The primers for *Mpl* were 5'-AACCCGGT ATGTGTGCCAG-3' (forward) and 5'-AGTTCATGCCTCAGGAAGTCA-3' (reverse). The primers for *Cxcl12* were 5'-AGGTTCTTATTTCACGGCTTGT-3' (forward) and 5'-TGGGTGCTGAGACCTTTGAT-3' (reverse). The primers for *Gapdh* were 5'-AGGTCGGTGTGAACGGATTTG-3' (forward) and 5'-GGGG TCGTTGATGGCAACA-3' (reverse). *Gapdh* was used as the reference gene for qRT-PCR analysis.

## Transwell migration

Transmigration assays were performed on a transwell with a pore size of 5 µm (Biofil). CXCR4$^{low}$ MKs or CXCR4$^{high}$ MKs from bone marrow were sorted (5000 cells per well) from control mice and added to the lower chamber with 600 µl IMDM (Thermo Fisher) plus 10% FBS (Gibco). Peripheral blood cells were collected as described in the 'Flow cytometry and cell sorting' section. 6×10$^5$ peripheral blood

cells were resuspended in 100 μl RPMI 1640 (Gibco) plus 10% FBS and added to the upper insert to continue for 2-hr incubation at 37 °C, 5% $CO_2$. Cells in the lower chamber were harvested, washed with PBS once, and resuspended with 100 μl PBS for staining and FACS counting.

## Phagocytosis

Bone-marrow-derived macrophages (BMDM) from C57BL/6 mice at 6–8 weeks of age were differentiated from bone marrow precursors with minor modifications (*Minutti et al., 2019*). In brief, bone marrow cells were isolated and propagated for 7 days in DMEM without sodium pyruvate or HEPES (Gibco), containing 20% FBS (Gibco), 30% supernatants of L929 conditioned media, and 1% Pen/Strep (Hyclone) at 37 °C. Macrophage phagocytosis assays were performed on a transwell plate with a pore size of 3 μm (Biofil) as described with modifications (*Sharif et al., 2014*). Briefly, attached cells were replated into 24-well plates, $5 \times 10^4$ cells per well, on glass coverslips for 24 hr culture. Then 5000 sorted CXCR4$^{low}$ MKs or CXCR4$^{high}$ MKs were added in the upper inserts and placed onto macrophages chambers for additional 16 hr incubation without or with 2 μg ml$^{-1}$ TNFα neutralizing antibody (R023, Sino Biological; 1:100) or 2 μg ml$^{-1}$ IL-6 neutralizing antibody (MP5-20F3, Biolegend) at 37 °C, 5% $CO_2$. The upper inserts were discarded and macrophages were washed with PBS without antibiotics and incubated with $10^5$ GFP-labeled *E. coli* for 2 hr at 37 °C, 5% $CO_2$. Cells were washed three times with PBS and incubated with DMEM without sodium pyruvate or HEPES (Gibco) with gentamycin (50 μg ml$^{-1}$) for 30 min at 37 °C, 5% $CO_2$ to remove adherent bacteria. Cells were then detected by flow cytometry or fixed by cold methanol for 15 min and blocked with 10% BSA overnight, followed by incubation with F4/80 (BM8, eBioscience; 1:100) for 2 hr at room temperature before being quantified using a spinning disk confocal microscope (Dragonfly, Andor).

For neutrophil phagocytosis, CD11b$^+$ Gr1$^+$ Ly6c$^-$ neutrophils were sorted from the spleen and propagated in RPMI 1640 (Gibco) containing 10% FBS. Neutrophil phagocytosis was performed as described in macrophage phagocytosis assay, except cells were sedimented for 30 min and fixed on glass coverslips after incubated with GFP-*E. coli* and gentamycin. The capacity of phagocytosis was evaluated by flow cytometry or by fluorescence intensity of GFP-*E. coli.* using the confocal microscope within macrophages and neutrophils.

For megakaryocyte phagocytosis, CXCR4$^{low}$ and CXCR4$^{high}$ MKs were sorted from the bone marrow and propagated in RPMI 1640 (Gibco) containing 10% FBS without antibiotics and incubated with $10^5$ GFP-labeled *E. coli* for 2 hr at 37 °C, 5% $CO_2$. Cells were washed three times with PBS and incubated with DMEM without sodium pyruvate or HEPES (Gibco) with gentamycin (50 μg ml$^{-1}$) for 30 min at 37 °C, 5% $CO_2$ to remove adherent bacteria. Cells were then detected by flow cytometry.

## Bone marrow ex vivo live imaging

*Pf4$^{Cre+}$*; *Rosa26$^{fs-tdTomato+/-}$* mice were infected with *L. monocytogenes* for 24 hr. FITC-Dextran (average mol wt 2000000, Sigma-Aldrich) was injected intravenously at a dose of 1.25 mg per mouse before being sacrificed. The ends of the femur below the end of the marrow cavity were cut. The bone marrow plug was gently flushed out of the end of the bone with a 21-gauge blunt needle not to break up the marrow plug. Bone marrow was flushed integrally and fixed onto a glass slide in a chamber, rinsed with RPMI 1640 (Gibco), and covered slightly with a coverslip. The integrity of the vascular structure in the bone marrow was observed and warranted through FITC-Dextran inflorescence before capturing images. Confocal images were obtained every minute on the spinning-disk confocal microscope (Dragonfly, Andor) and analyzed using Imaris 9.0 software (Oxford Instruments).

## In vitro MK culture, MK size, and proplatelet formation measurement

MKs were sorted using a cell sorter (MoFlo Astrios, Beckman Coulter) and cultured in 24-well plates in SFEM (Stem Cell Technologies) plus 100 ng ml$^{-1}$ mTPO (Novoprotein) and 1% Pen/Strep (Hyclone), and incubated at 37 °C, 5% $CO_2$ for 4 days. Images were taken by a Nikon Ts2R microscope equipped with a Nikon DS-Ri2 camera. Cell size and proplatelet formation were measured on day 3 or day 5 post-cultured, respectively, using Nikon NIS-Elements BR.

## Bone marrow transfer experiments

*Pf4$^{Cre}$* mice were mated with the *Rosa26$^{fs-tdTomato}$* line to generate *Pf4$^{Cre+}$*; *Rosa26$^{fs-tdTomato+/-}$* mice. tdTomato$^+$ MKs were isolated from *Pf4$^{Cre+}$*; *Rosa26$^{fs-tdTomato+/-}$* mice. Six- to 8-week-old recipient mice were

pre-treated with PBS or 2500 CFUs of *L. monocytogenes* as previously described 1 day before cell perfusion. $1 \times 10^5$ tdTomato$^+$ MKs were sorted and intravenously injected into control or *L. monocytogenes* infected mice. tdTomato$^+$ MKs were detected in lungs with immunostaining 2 days after cell perfusion.

mtdTomato$^+$ bone marrow cells were isolated from *Pf4$^{Cre-}$*; *Rosa26$^{fs-mTmG+/-}$* mice. $1 \times 10^6$ mtdTomato$^+$ bone marrow cells were intravenously injected into control or one-day-*L. monocytogenes* infected mice. mtdTomato$^+$ MKs were detected in bone marrow, liver, and spleen 2 days after cell perfusion.

For in vivo CXCR4$^{high}$ MK function assay in MK ablation mice, DT was intraperitoneally injected every day for 5 days. On the second and fourth days, $2 \times 10^5$ sorted wild-type CXCR4$^{high}$ MKs or CXCR4$^{low}$ MKs were intravenously injected into indicated groups. PBS or 2500 CFUs of *L. monocytogenes* as previously described were injected intravenously on the third day. Spleen and liver were harvested 3 days after infection to determine the bacterial burdens as described.

### T cell reactivation in vitro

Splenocytes ($1 \times 10^6$ cells well$^{-1}$) from control or MK ablated mice after 7 days *L.m.*-OVA infection were re-stimulated for 4 hr in vitro with OVA peptide (10 μM) in the presence of Brefeldin-A (BFA, 10 μg ml$^{-1}$). Activated T cells were then analyzed by a flow cytometer.

For MK-induced T cell activation, $3 \times 10^4$ MK subpopulations for each sample were sorted and co-cultured with 100 μg ml$^{-1}$ soluble full-length OVA for 24 hr, then co-cultured with $6 \times 10^4$ OT-I CD8$^+$ T cells or B3Z T cells (*Karttunen et al., 1992*) for 48 hr at 37 °C in a 5% CO$_2$ incubator as described (*Zufferey et al., 2017*). OT-I T cell activation was detected by measuring intracellular IL-2 levels. B3Z T cell activation was detected using β-galactosidase Assay Kit (RG0036, Beyotime). Bone marrow-derived dendritic cells (DCs) were adopted as positive controls for T cell activation assay. To obtain bone marrow-derived DCs, isolated bone marrow cells were cultured in RPMI 1640 with 10 ng ml$^{-1}$ of GM-CSF and 10 ng ml$^{-1}$ of IL-4 as described (*Roney, 2019*).

### Computational modeling of random myeloid cell localization

We have performed randomized simulations as in previous reports (*Bruns et al., 2014*; *Jiang et al., 2018*) in Python. Images of a 400 μm × 400 μm bone marrow region with CXCR4$^{high}$ and CXCR4$^{low}$ MKs, in which background staining was removed, were used to generate MKs onto which 200 myeloid cells were randomly placed, consistent with an average density of 200 myeloid cells per field. Each simulated run placed 200 random myeloid cells (mean diameter 5 μm) was repeated 500 times. The shortest Euclidean distance was calculated for each myeloid cell to CXCR4$^{high}$ or CXCR4$^{low}$ MKs. Random and observed distance distributions were compared using the modified nonparametric two-dimensional (2D) KS test as described (*Bruns et al., 2014*; *Jiang et al., 2018*).

### Statistical analyses

Data are presented as means ± s.e.m or presented medians, first and third quartiles. For phagocytosis assay and MK size measurement, data were analyzed by a one-dimensional KS test. Differences were considered statistically significant if $p < 0.05$. For the comparison of three-dimensional distances, a two-dimensional KS test was used. The difference was considered statistically significant if $p < 0.05$. For multiple comparisons analysis, data were analyzed by repeated-measures one-way analysis of variance (ANOVA) followed by Dunnett's test. Differences were considered statistically significant if $p < 0.05$. ǂ $p < 0.05$, ♯ $p < 0.01$, ♯♯ $p < 0.001$, n.s., not significant. For pairs of measurements, data were analyzed by paired Student's *t*-test. Differences were considered statistically significant if $p < 0.05$. # $p < 0.05$, ## $p < 0.01$, ### $p < 0.001$, n.s., not significant. For other experiments except for scRNA-seq analysis, data were analyzed by a two-tailed Student's *t*-test. Differences were considered statistically significant if $p < 0.05$. * $p < 0.05$, ** $p < 0.01$, *** $p < 0.001$, n.s., not significant.

## Acknowledgements

We thank the National Key Research and Development Program of China (2018YFA0107200), the National Natural Science Foundation of China (82170112, 81900101), The Key Research and Development Program of Guangdong Province (2019B020234002), China Postdoctoral Science Foundation (2021M693614), Guangdong Innovative and Entrepreneurial Research Team Program (2016ZT06S029,

2019ZT08Y485), Sanming Project of Medicine in Shenzhen (No.SZSM201911004) for generous support.

## Additional information

### Funding

| Funder | Grant reference number | Author |
|---|---|---|
| National Key Research and Development Program of China | 2018YFA0107200 | Meng Zhao |
| National Natural Science Foundation of China | 82170112 | Meng Zhao |
| National Natural Science Foundation of China | 81900101 | Jin Wang |
| Key Research and Development Program of Guangdong Province | 2019B020234002 | Meng Zhao |
| Guangdong Innovative and Entrepreneurial Research Team Program | 2016ZT06S029 | Meng Zhao |
| Guangdong Innovative and Entrepreneurial Research Team Program | 2019ZT08Y485 | Linjia Jiang |
| Sanming Project of Medicine in Shenzhen | SZSM201911004 | Meng Zhao |
| China Postdoctoral Science Foundation | 2021M693614 | Jin Wang |

The funders had no role in study design, data collection and interpretation, or the decision to submit the work for publication.

### Author contributions

Jin Wang, Resources, Data curation, Formal analysis, Funding acquisition, Validation, Investigation, Visualization, Methodology, Writing - original draft, Project administration, Writing - review and editing; Jiayi Xie, Data curation, Formal analysis, Validation, Investigation, Visualization, Methodology, Writing - review and editing; Daosong Wang, Data curation, Investigation, Visualization, Methodology; Xue Han, Minqi Chen, Data curation; Guojun Shi, Linjia Jiang, Supervision; Meng Zhao, Conceptualization, Supervision, Funding acquisition, Writing - original draft, Project administration, Writing - review and editing

### Author ORCIDs

Jin Wang ⓘ http://orcid.org/0000-0002-4924-1716
Jiayi Xie ⓘ http://orcid.org/0000-0001-8977-6654
Daosong Wang ⓘ http://orcid.org/0000-0002-4786-6202
Linjia Jiang ⓘ http://orcid.org/0000-0001-8854-2610
Meng Zhao ⓘ http://orcid.org/0000-0001-7909-7594

### Ethics

All animal experiments were performed according to protocols approved by the Institutional Animal Care and Use Committee.

### Decision letter and Author response

Decision letter https://doi.org/10.7554/eLife.78662.sa1
Author response https://doi.org/10.7554/eLife.78662.sa2

# Additional files

## Supplementary files
• MDAR checklist

## Data availability

The scRNA-seq data generated in this study are deposited in GEO (GSE168224, https://www.ncbi.nlm.nih.gov/geo/query/acc.cgi?acc=GSE168224). The code used in the study can be accessed at GitHub (https://github.com/JYCathyXie/MK_infection, copy archived at swh:1:rev:687f151f79a79ad2091e3dc2c5561fc8b4bb347a).

The following dataset was generated:

| Author(s) | Year | Dataset title | Dataset URL | Database and Identifier |
|---|---|---|---|---|
| Zhao M, Wang J, Xie J, Wang D, Han X | 2022 | Megakaryocyte derived immunoregulatory cells regulate host-defense immunity against bacterial pathogens | https://www.ncbi.nlm.nih.gov/geo/query/acc.cgi?acc=GSE168224 | NCBI Gene Expression Omnibus, GSE168224 |

The following previously published datasets were used:

| Author(s) | Year | Dataset title | Dataset URL | Database and Identifier |
|---|---|---|---|---|
| Yeung AK, Villacorta-Martin C, Murphy GJ | 2021 | Single Cell Transcriptomic Analysis of Lung and Hematopoietic Megakaryocytes from Embryonic and Adult Mice | https://www.ncbi.nlm.nih.gov/geo/query/acc.cgi?acc=GSE152574 | NCBI Gene Expression Omnibus, GSE152574 |
| Pariser DN, Hilt ZT, Ture SK, Blick-Nitko SK, Looney MR, Cleary SJ, Roman-Pagan E, Saunders J, Georas SN, Veazey J, Madera F, Santos LT, Arne A, Huynh NT, Livada AC, Guerrero-Martin SM, Lyons C, Metcalf-Pate KA, McGrath KE, Palis J, Morrell CN | 2020 | Lung Megakaryocytes are Immune Modulatory Cells | https://www.ncbi.nlm.nih.gov/geo/query/acc.cgi?acc=GSE158358 | NCBI Gene Expression Omnibus, GSE158358 |
| Xie X, Shi Q, Wu P, Zhang X | 2020 | Single-cell transcriptome profiling reveals neutrophil heterogeneity in homeostasis and infection | https://www.ncbi.nlm.nih.gov/geo/query/acc.cgi?acc=GSE137540 | NCBI Gene Expression Omnibus, GSE137540 |
| Hamey FK, Lau WW, Diamanti E, Göttgens B, Dahlin JS | 2020 | Single-cell RNA sequencing of basophils from mouse bone marrow | https://www.ncbi.nlm.nih.gov/geo/query/acc.cgi?acc=GSE128074 | NCBI Gene Expression Omnibus, GSE128074 |
| Tabula Muris Consortium | 2019 | Expression profiling by high throughput sequencing | https://www.ncbi.nlm.nih.gov/geo/query/acc.cgi?acc=GSE132042 | NCBI Gene Expression Omnibus, GSE132042 |

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
