## [Editor Report]

The manuscript by Wang and colleagues studies the heterogeneity of megakaryocytes using single-cell RNA-seq and identifies a subpopulation of CXCR4-high megakaryocytes with immune modulatory roles. Functional studies in this paper show that this subpopulation of megakaryocytes promotes bacterial phagocytosis by macrophages and neutrophils. The compelling evidence for these findings in the manuscript and the fundamental significance of identifying mechanisms by which megakaryocyte subpopulations can impact host defense make this work of interest to researchers in the fields of immunology, hematopoiesis and megakaryocyte biology.

---

## [Decision Letter]

**Decision letter after peer review:**

Thank you for submitting your article "Megakaryocyte derived immune-stimulating cells regulate host-defense immunity against bacterial pathogens" for consideration by *eLife*. Your article has been reviewed by 3 peer reviewers, including Jalees Rehman as Reviewing Editor and Reviewer #1, and the evaluation has been overseen by Jos van der Meer as the Senior Editor.

Essential revisions:

1. Please clarify whether these are Megakaryocyte-derived immune cells (MDICs) or simply a subpopulation of megakaryocytes (MK5). The presented data does not provide any evidence for the "derivation" process such as differentiation factors that convert the megakaryocyte subpopulation MK5 into a different "immune cell". Therefore, it is requested to change the MDIC term and instead use an expression that indicates that these cells are a subset of megakaryocytes.

2. Please perform studies to ascertain the phagocytosis potential of this megakaryocyte subset. Are they capable of direct bacterial phagocytosis in the absence of other immune cells or do they only enhance bacterial phagocytosis by other immune cells?

3. Does this CXCR4 high megakaryocyte subset overlap with other megakaryocyte subsets demonstrating immune cell functions that have been recently reported such as antigen processing or antigen presentation? Assessing their antigen processing or antigen presentation capacity would help define the roles of this MK subset in host defense.

4. Does the CXCR4 high megakaryocyte subset also generate platelets or is their platelet generation function diminished compared to other megakaryocyte subsets, thus indicating that they are more specialized for immune functions.

5. How is this CXCR4 subset generated and what is their fate after infusion? Does CXCR4 get upregulated during infection on non-CXCR4 megakaryocytes? This would suggest that they are not terminally programmed to be CXCR4 high but that other megakaryocyte subsets can become MK5/CXCR4 high and take on such host defense functions.

6. Conclusions and generalizations need to be adjusted based on the comments of the reviewers as provided below. One specific example is the need to moderate claims about the role of megakaryocytes in adaptive immunity as this was not addressed in depth. Other examples of conclusions that need to be moderated are also listed by the reviewers in their specific comments.

7. Some technical details need to be addressed such as clarifying flow cytometry gating strategies or describing the new imaging approach. Description of the imaging approach and objective quantification of imaging data is essential to increase the robustness of the conclusions. The goal is for readers to be able to understand and replicate the experiments. More protein level quantifications of cytokines using flow cytometry or ELISA are also required to justify several of the conclusions.

*Reviewer #1 (Recommendations for the authors):*

Specific suggestions to include in the manuscript would include:

1. Perform antigen processing assays with CXCR4 high and CXCR4 low MKs to then assess whether this is the subset that drives the adaptive immune modulatory function

2. Use CXCR4 low cells and expose them ex vivo to inflammatory or infectious stimuli to assess whether they can become CXCR4 high. This would assess the plasticity of the MK subsets.

*Reviewer #2 (Recommendations for the authors):*

Previous studies suggested that immune MKs are capable of phagocytosis themselves and not just supporting immune cells. Are MDIC cells capable of direct bacterial phagocytosis?

In Figure 2, it's unclear if the myeloid cell-CXCR4high MK association is simply a result of the expansion of myeloid cells upon infection. Is this association observed in homeostasis?

What happens to CXCR4 expression on MKs upon infection? It would be important to understand if CXCR4low MKs are "reprogrammed" into CXCR4high MKs upon infection.

Other technical concerns:

– Elisa or flow cytometry analyses would be more accurate to evaluate TNFα, IL-6, and CXCL12 levels.

– The authors describe a new method of "bone marrow live imaging", however it's unclear how imaging of flushed bone marrow plugs ex vivo accurately reflects live imaging.

– It's unclear the gating strategy used to quantify bone marrow, liver and spleen myeloid cell populations in Figure 3 as some of these populations rely on tissue specific markers.

– In Figure 4 supplemental 3, it's unusual to present CXCL12 levels as mRNA expression from total cells in the tissue, and it's even more surprising that the authors can detect accurate levels in organs such as the bone marrow or spleen since only small subsets of cells express CXCL12 in the tissue.

– Small concern, in Figure 4G the results are presented in different formats alternating between absolute number quantification or percentages.

*Reviewer #3 (Recommendations for the authors):*

There are a number of specific things the authors can do to improve this manuscript.

1. The term MDIC should be reconsidered – especially the meaning of the D – are they Mk derived, or a subset of Mks? Since seem to be a subset, the derived is not really appropriate.

2. The authors indicate that the MDIC enhance the phagocytosis capacity of other cells – but do they also phagocytose bacteria? Is this contact dependent or independent to better show is a secreted product or not.

3. The DT model ablates both Mks and platelets. Therefore many of the changes in bacteria and immune cells could be platelet driven, not Mk or MDIC driven. This must be reconciled or the conclusions drastically restated in Figure 3.

4. Where do the infused MDIC go? Marrow, spleen, lung or all of the above? Do they make platelets and if so better or worse than Mk1 or Mk2 populations?

5. Do any of the lung migrated Mk become extravascular lung Mks, as 2/3 of lung Mks are extravascular? Or do they stay intravascular platelet producing Mks?

---

## [Author Response]

Essential Revisions (for the authors):1. Please clarify whether these are Megakaryocyte-derived immune cells (MDICs) or simply a subpopulation of megakaryocytes (MK5). The presented data does not provide any evidence for the "derivation" process such as differentiation factors that convert the megakaryocyte subpopulation MK5 into a different "immune cell". Therefore, it is requested to change the MDIC term and instead use an expression that indicates that these cells are a subset of megakaryocytes.

Thanks for the suggestion. We have changed the MDICs term to CXCR4^high^ MKs.

2. Please perform studies to ascertain the phagocytosis potential of this megakaryocyte subset. Are they capable of direct bacterial phagocytosis in the absence of other immune cells or do they only enhance bacterial phagocytosis by other immune cells?

Our revised data showed that CXCR4^high^ MKs were capable of bacterial phagocytosis (Revised Figure 3F).

3. Does this CXCR4 high megakaryocyte subset overlap with other megakaryocyte subsets demonstrating immune cell functions that have been recently reported such as antigen processing or antigen presentation? Assessing their antigen processing or antigen presentation capacity would help define the roles of this MK subset in host defense.

Our revised data showed that CXCR4^high^ MKs were able to phagocytose bacteria (Revised Figure 3F), and process and present ovalbumin (OVA) antigens on their cell surface (Revised Figure 3G) to activate CD8^+^ OT-I T cells (Revised Figure 3H) and B3Z T cells (Revised Figure 3-S2), a T cell hybridoma which expresses TCR that specifically recognizes OVA. These revised data showed that CXCR4^high^ MKs had antigen processing and antigen presentation capacity.

4. Does the CXCR4 high megakaryocyte subset also generate platelets or is their platelet generation function diminished compared to other megakaryocyte subsets, thus indicating that they are more specialized for immune functions.

Our revised data showed that CXCR4^high^ MKs also generated platelets but with lower efficiency compared to CXCR4^low^ MKs (Revised Figure 1-S6D), suggesting CXCR4^high^ MKs might be specialized for immune functions.

5. How is this CXCR4 subset generated and what is their fate after infusion? Does CXCR4 get upregulated during infection on non-CXCR4 megakaryocytes? This would suggest that they are not terminally programmed to be CXCR4 high but that other megakaryocyte subsets can become MK5/CXCR4 high and take on such host defense functions.

Our revised data showed that inflammatory treatment, including interferon γ, LPS, and *L. monocytogenes* could not increase CXCR4 expression in CXCR4^low^ MKs (Revised Figure 4H and Figure 4-S4D). This experiment suggested that CXCR4^high^ MKs might not be reprogramed from CXCR4^low^ MKs. Furthermore, our HSPC tracing experiment showed that CXCR4^high^ MKs were generated from HSPCs as efficiently as CXCR4^low^ MKs during the acute inflammation-induced emergency megakaryopoiesis (Revised Figure 5E-G).

6. Conclusions and generalizations need to be adjusted based on the comments of the reviewers as provided below. One specific example is the need to moderate claims about the role of megakaryocytes in adaptive immunity as this was not addressed in depth. Other examples of conclusions that need to be moderated are also listed by the reviewers in their specific comments.

We have removed the conclusion about adaptive immunity as suggested.

7. Some technical details need to be addressed such as clarifying flow cytometry gating strategies or describing the new imaging approach. Description of the imaging approach and objective quantification of imaging data is essential to increase the robustness of the conclusions. The goal is for readers to be able to understand and replicate the experiments. More protein level quantifications of cytokines using flow cytometry or ELISA are also required to justify several of the conclusions.

In this revision, we have added flow cytometry gating strategies (Revised Figure 3-S1C, E, F and Figure 4-S4G, H), and the description for the ex vivo live imaging approach in the method section (Line 7-18, Page 26). We have also performed flow cytometry assays to confirm the imaging assays for bacterial phagocytosis assays, quantification of MKs and cytokine expression (Revised Figure 2J, M, N, Figure 2-S2A, B, Figure 4I, O and Figure 4-S4I).

Reviewer #1 (Recommendations for the authors):Specific suggestions to include in the manuscript would include:1. Perform antigen processing assays with CXCR4 high and CXCR4 low MKs to then assess whether this is the subset that drives the adaptive immune modulatory function

Thanks for the suggested experiment. Our revised data showed that CXCR4^high^ MKs, but not CXCR4^low^ MKs, were able to process and present ovalbumin (OVA) antigens on their cell surface (Revised Figure 3G) and activate T cells (Revised Figure 3H and Figure 3-S2).

2. Use CXCR4 low cells and expose them ex vivo to inflammatory or infectious stimuli to assess whether they can become CXCR4 high. This would assess the plasticity of the MK subsets.

Our revised data showed that inflammatory treatment, including interferon γ, LPS, and *L. monocytogenes* could not increase CXCR4 expression in CXCR4^low^ MKs (Revised Figure 4H and Figure 4-S4D). This experiment suggested that CXCR4^high^ MKs might not be reprogramed from CXCR4^low^ MKs, although we cannot entirely exclude the plasticity of MK subsets. We have included a brief discussion in the current version (Line 20-22, Page14).

Reviewer #2 (Recommendations for the authors):Previous studies suggested that immune MKs are capable of phagocytosis themselves and not just supporting immune cells. Are MDIC cells capable of direct bacterial phagocytosis?

Our revised data showed that MDICs (now referred to as CXCR4^high^ MKs) were capable of phagocytosis for bacteria (Revised Figure 3F).

In Figure 2, it's unclear if the myeloid cell-CXCR4high MK association is simply a result of the expansion of myeloid cells upon infection. Is this association observed in homeostasis?

In this revision, we showed that CXCR4^high^ MKs were significantly associated with myeloid cells upon infection compared with randomly distributed spots in the mathematical modeling (Revised Figure 2B-C). This may not be due to the expansion of myeloid cells, as the association between CXCR4^low^ MKs and myeloid cells was not significant compared to randomly distributed spots (Revised Figure 2C). Furthermore, we did not observe the association between CXCR4^high^ MKs and myeloid cells during homeostasis (Revised Figure 2-S1C, D).

What happens to CXCR4 expression on MKs upon infection? It would be important to understand if CXCR4low MKs are "reprogrammed" into CXCR4high MKs upon infection.

Thanks for the suggested experiment. Our revised data showed that inflammatory treatment, including Interferon γ, LPS, and *L. monocytogenes* could not increase CXCR4 expression in CXCR4^low^ MKs (Revised Figure 4H and Figure 4-S4D). This experiment suggested that CXCR4^high^ MKs might not be reprogramed from CXCR4^low^ MKs, although we cannot entirely exclude the plasticity of MK subsets. We have included a brief discussion in the current version (Line 20-22, Page 14).

Other technical concerns:– Elisa or flow cytometry analyses would be more accurate to evaluate TNFα, IL-6, and CXCL12 levels.

Thanks for the suggested experiments. In this revision, we have further evaluated the expression of TNFα and IL-6 by flow cytometry, which consistently showed that CXCR4^high^ MKs had higher expression levels of TNFα and IL-6 than CXCR4^low^ MKs (Revised Figure 2J). We have also adapted *Cxcl12^DsRed^* reporter mouse line to show that CXCL12 was upregulated in the liver and spleen but reduced in bone marrow upon infection by flow cytometry assays (Revised Figure 4-S3E-H)*.*

– The authors describe a new method of "bone marrow live imaging", however it's unclear how imaging of flushed bone marrow plugs ex vivo accurately reflects live imaging.

Sorry for the inaccurate description. This an ex vivo imaging approach, which is developed before for the real-time imaging in the bone (Xie et al., Nature 2009). We have described this ex vivo imaging approach in the result and method section (Line 7-18, Page 26).

– It's unclear the gating strategy used to quantify bone marrow, liver and spleen myeloid cell populations in Figure 3 as some of these populations rely on tissue specific markers.

Thanks for pointing out this technical issue. To avoid inconsistency, we analyzed the CD11b^+^ Ly6c^+^ monocytes, CD11b^+^F4/80^+^Ly6c^–^ macrophages, CD11c^+^CD11b^+^ dendritic cells, and CD11b^+^Ly6c^–^Gr-1^+^ neutrophils from bone marrow, liver, and spleen. The gating strategy has been used in previous studies (Krenkel et al., Gut 2020; Swirski et al., Science 2009). We have also included the representative gating strategies for myeloid cells (Revised Figure 3-S1C, E, F).

– In Figure 4 supplemental 3, it's unusual to present CXCL12 levels as mRNA expression from total cells in the tissue, and it's even more surprising that the authors can detect accurate levels in organs such as the bone marrow or spleen since only small subsets of cells express CXCL12 in the tissue.

We agree with this argument. In this revision, we employed a *Cxcl12^DsRed^* reporter mouse line, and further confirmed that CXCL12 was upregulated in the liver and spleen but reduced in bone marrow upon infection (Revised Figure 4-S3E-H).

– Small concern, in Figure 4G the results are presented in different formats alternating between absolute number quantification or percentages.

Thanks for the suggestion. We presented these results in percentage in the current version (Revised Figure 4G).

Reviewer #3 (Recommendations for the authors):There are a number of specific things the authors can do to improve this manuscript.1. The term MDIC should be reconsidered – especially the meaning of the D – are they Mk derived, or a subset of Mks? Since seem to be a subset, the derived is not really appropriate.

We agree with this argument. We have changed the MDIC term to CXCR4^high^ MKs in the current version as suggested.

2. The authors indicate that the MDIC enhance the phagocytosis capacity of other cells – but do they also phagocytose bacteria? Is this contact dependent or independent to better show is a secreted product or not.

Thanks for pointing out this issue. Our revised data showed that CXCR4^high^ MKs were capable of bacterial phagocytosis, although the efficiency was relatively lower compared to neutrophils (Revised Figure 3F). Furthermore, CXCR4^high^ MKs promoted the phagocytosis capacity of neutrophils and macrophages in the upper inserts of transwell plates (Revised Figure 2E-H and Figure 2-S2A, B), which indicated that CXCR4^high^ MKs promote the phagocytosis capacity of other cells through a cell-contact independent manner. By using neutralizing antibodies, we also showed that CXCR4^high^ MKs promote the phagocytosis of immune cells partially through secreting IL-6 and TNFα (Revised Figure 2K-N).

3. The DT model ablates both Mks and platelets. Therefore many of the changes in bacteria and immune cells could be platelet driven, not Mk or MDIC driven. This must be reconciled or the conclusions drastically restated in Figure 3.

We agree with this argument. Our infusion rescue with CXCR4^high^ MKs did not fully rescue the host-defense responses in MK ablated mice, which might be partially due to the reduced platelets, which are known for immune responses. We have discussed this possibility in the current version (Line 16-17, Page 9).

4. Where do the infused MDIC go? Marrow, spleen, lung or all of the above? Do they make platelets and if so better or worse than Mk1 or Mk2 populations?

The infused MDICs (now referred to as CXCR4^high^ MKs) preferably migrated to the liver and spleen; conversely, the infused CXCR4^low^ MKs preferably migrated to the bone marrow (Revised Figure 4J-O). Furthermore, CXCR4^high^ MKs produced fewer platelets than CXCR4^low^ MKs, which enriched MK1 and MK2 populations (Revised Figure 1-S6D).

5. Do any of the lung migrated Mk become extravascular lung Mks, as 2/3 of lung Mks are extravascular? Or do they stay intravascular platelet producing Mks?

Thanks for the suggested experiments. Our revised data showed that the lung migrated MKs were extravascular lung MKs (Revised Figure 4-S6).

References

Beaulieu, L. M., Lin, E., Morin, K. M., Tanriverdi, K., and Freedman, J. E. (2011). Regulatory effects of TLR2 on megakaryocytic cell function. *Blood*, *117*(22), 5963-5974. https://doi.org/10.1182/blood-2010-09-304949

Krenkel, O., Hundertmark, J., Abdallah, A. T., Kohlhepp, M., Puengel, T., Roth, T., Branco, D. P. P., Mossanen, J. C., Luedde, T., Trautwein, C., Costa, I. G., and Tacke, F. (2020). Myeloid cells in liver and bone marrow acquire a functionally distinct inflammatory phenotype during obesity-related steatohepatitis. *Gut*, *69*(3), 551-563. https://doi.org/10.1136/gutjnl-2019-318382

Markovic, B., Wu, Z., Chesterman, C. N., and Chong, B. H. (1995). Quantitation of soluble and membrane-bound Fc γ RIIA (CD32A) mRNA in platelets and megakaryoblastic cell line (Meg-01). *Br J Haematol*, *91*(1), 37-42. https://doi.org/10.1111/j.1365-2141.1995.tb05241.x

Navarro, S., Debili, N., Le Couedic, J. P., Klein, B., Breton-Gorius, J., Doly, J., and Vainchenker, W. (1991). Interleukin-6 and its receptor are expressed by human megakaryocytes: in vitro effects on proliferation and endoreplication. *Blood*, *77*(3), 461-471. https://www.ncbi.nlm.nih.gov/pubmed/1991163

Negrotto, S., C, J. D. G., Lapponi, M. J., Etulain, J., Rivadeneyra, L., Pozner, R. G., Gomez, R. M., and Schattner, M. (2011). Expression and functionality of type I interferon receptor in the megakaryocytic lineage. *J Thromb Haemost*, *9*(12), 2477-2485. https://doi.org/10.1111/j.1538-7836.2011.04530.x

Swirski, F. K., Nahrendorf, M., Etzrodt, M., Wildgruber, M., Cortez-Retamozo, V., Panizzi, P., Figueiredo, J. L., Kohler, R. H., Chudnovskiy, A., Waterman, P., Aikawa, E., Mempel, T. R., Libby, P., Weissleder, R., and Pittet, M. J. (2009). Identification of splenic reservoir monocytes and their deployment to inflammatory sites. *Science*, *325*(5940), 612-616. https://doi.org/10.1126/science.1175202

Ward, J. R., Bingle, L., Judge, H. M., Brown, S. B., Storey, R. F., Whyte, M. K., Dower, S. K., Buttle, D. J., and Sabroe, I. (2005). Agonists of toll-like receptor (TLR)2 and TLR4 are unable to modulate platelet activation by adenosine diphosphate and platelet activating factor. *Thromb Haemost*, *94*(4), 831-838. https://www.ncbi.nlm.nih.gov/pubmed/16270639

Xie, Y., Yin, T., Wiegraebe, W., He, X. C., Miller, D., Stark, D., Perko, K., Alexander, R., Schwartz, J., Grindley, J. C., Park, J., Haug, J. S., Wunderlich, J. P., Li, H., Zhang, S., Johnson, T., Feldman, R. A., and Li, L. (2009). Detection of functional haematopoietic stem cell niche using real-time imaging. *Nature*, *457*(7225), 97-101. https://doi.org/10.1038/nature07639

Yang, M., Li, K., Chui, C. M., Yuen, P. M., Chan, P. K., Chuen, C. K., Li, C. K., and Fok, T. F. (2000). Expression of interleukin (IL) 1 type I and type II receptors in megakaryocytic cells and enhancing effects of IL-1beta on megakaryocytopoiesis and NF-E2 expression. *Br J Haematol*, *111*(1), 371-380. https://doi.org/10.1046/j.1365-2141.2000.02340.x